# DIVIDE AND CONQUER: PROVABLY UNVEILING THE PARETO FRONT WITH MULTI-OBJECTIVE REINFORCEMENT LEARNING

## ABSTRACT

We introduce a novel algorithm for learning the Pareto front in multi-objective Markov decision processes. Our algorithm decomposes learning the Pareto front into a sequence of single-objective problems, each of which is solved by an oracle and leads to a non-dominated solution. We propose a procedure to select the single-objective problems such that each iteration monotonically decreases the objective space that possibly still contains Pareto optimal solutions. The final algorithm is proven to converge to the Pareto front and provides an upper bound on the distance to undiscovered non-dominated policies in each iteration. We introduce several practical designs of the required oracle by extending single-objective reinforcement learning algorithms. When evaluating our algorithm with these oracles on benchmark environments, we find that it leads to a close approximation of the true Pareto front. By leveraging problem-specific single-objective solvers, our approach holds promise for applications beyond multi-objective reinforcement learning, such as in pathfinding and optimisation.

## 1 INTRODUCTION

In many real-world decision-making problems, there are multiple and conflicting objectives to optimise for. Formulating effective environmental policies, for example, involves a complex trade-off between economic and climate metrics (Cui et al., 2017). Given that there is often no single policy that maximises all objectives simultaneously, it is common to compute a set of candidate optimal policies. This set is subsequently presented to a decision maker who selects their preferred policy from it. This approach has multiple advantages such as systematically exploring the space of trade-offs and empowering a decision-maker to examine the outcomes of their decisions with respect to their objectives (Hayes et al., 2022). We consider the problem of learning such a set of policies with reinforcement learning in the context of multi-objective Markov decision processes (MOMDPs).

Depending on the specific setting, the set of policies allowed and prior knowledge about the decision maker, it is possible to define different types of solution sets for this purpose (Roijers & Whiteson, 2017). A solution set that is often considered in MOMDPs is the convex hull (Yang et al., 2019; Alegre et al., 2023), which has been shown to be optimal whenever decision-makers have linear utility functions or when stochastic policies are allowed. When the preferences over objectives may be non-linear and deterministic policies are required for interpretability or safety reasons, it is appropriate to learn a Pareto front of policies (Hayes et al., 2022).

In multi-objective optimisation, one of the most successful paradigms for computing a Pareto front is to decompose the multi-objective problem into single-objective problems which may be solved and combined at a later stage (Santiago et al., 2014). Such algorithms can usually be efficiently parallelised and scale well with additional computing power. Moreover, it is often possible to adapt well-known single-objective methods to solve the decomposed problems, thereby creating a strong connection between progress in multi-objective and single-objective methods.

In the same spirit, we propose a novel multi-objective reinforcement learning (MORL) algorithm, Iterated Pareto Referent Optimisation (IPRO), that learns the Pareto front by decomposing the problem into a sequence of single-objective problems. IPRO iteratively proposes reference points to a Pareto oracle and uses the solution returned by the oracle to trim sections from the search space.

To the best of our knowledge, this is the first algorithm that comes with an upper bound on the approximation error to the true Pareto front at each iteration and guarantees convergence. We modify strong single-objective RL algorithms to function as Pareto oracles and evaluate IPRO on standard benchmarks. We find that it learns high-quality Pareto fronts that closely approximate the true Pareto front when known and otherwise are competitive to those produced by state-of-the-art methods. We summarise the core contributions of our algorithm relative to related work in Table 1.

| Algorithm | Convergence | Runtime guarantees | Preferences | MOMDPs |
|---|---|---|---|---|
| PCN (Reymond et al., 2022) | ✗ | ✗ | (Non)linear | Deterministic |
| PG-MORL (Xu et al., 2020) | ✗ | ✗ | Linear | General |
| Envelope (Yang et al., 2019) | ✓ | ✗ | Linear | General |
| GPI-LS (Alegre et al., 2023) | ✓ | ✓ | Linear | General |
| **IPRO (ours)** | ✓ | ✓ | (Non)linear | General |

Table 1: A summary of our contributions compared to related work.

**Related work**. When learning a single policy in MOMDPs, as is necessary for our Pareto oracles, conventional methods often adapt single-objective RL algorithms. For example, Siddique et al. (2020) extend DQN, A2C and PPO to learn a fair policy by optimising the generalised Gini index of the multi-objective expected returns. Reymond et al. (2023) extend this approach to general non-linear functions and establish a policy gradient theorem for this setting. Finally, when maximising a concave function of the expected returns, efficient methods exist which guarantee global convergence (Zhang et al., 2020; Zahavy et al., 2021; Geist et al., 2022).

To learn a Pareto front of policies in deterministic MOMDPs, Reymond et al. (2022) employ a single neural network to predict deterministic policies for various desired trade-off points. For scenarios involving stochastic policies, Xu et al. (2020) introduce an evolutionary learning algorithm tailored to continuous control settings. Their algorithm trains a population of policies and evolves this population using a prediction model, which estimates the expected improvement along each objective. Furthermore, Lu et al. (2023) demonstrate that for this setting it is possible to add a strongly concave term to the reward function and induce a sequence of single-objective problems with different weights. Closely related to our approach, Van Moffaert et al. (2013) learn a Pareto front of deterministic policies by decomposing the problem into a sequence of single-objective problems using the Chebyshev scalarisation function. However, their method has no theoretical guarantees and does not extend to settings with continuous state or action spaces.

Finally, when learning a convex hull, which is appropriate for decision-makers with linear utility functions, most techniques rely on the fact that the overall problem can be decomposed into single-objective problems where the scalar reward is a convex combination of the original reward vector (Yang et al., 2019; Alegre et al., 2023). The success of decomposition approaches in this setting, as well as for obtaining a Pareto front in multi-objective optimisation (Zhang & Li, 2007) further motivates their application to learning a Pareto front.

## 2 PRELIMINARIES

**Pareto dominance**. For two vectors $\mathbf{v}, \mathbf{v}' \in \mathbb{R}^d$ we say that $\mathbf{v}$ Pareto dominates $\mathbf{v}'$, denoted $\mathbf{v} \succ \mathbf{v}'$, when $\forall j \in \{1, \ldots, d\} : \mathbf{v}_j \geq \mathbf{v}'_j$ and $\mathbf{v}_j > \mathbf{v}'_j$ for at least one $j$. When dropping the strict condition, we write $\mathbf{v} \succeq \mathbf{v}'$. We say that $\mathbf{v}$ strictly Pareto dominates $\mathbf{v}'$, denoted $\mathbf{v} > \mathbf{v}'$ when $\forall j \in \{1, \ldots, d\} : \mathbf{v}_j > \mathbf{v}'_j$. When a vector is not pairwise (strictly) Pareto dominated, we say it is (strict) Pareto optimal. Finally, a vector is weakly Pareto optimal whenever there is no other vector that strictly Pareto dominates it.

In multi-objective decision-making, Pareto optimal vectors are especially relevant when considering decision-makers with monotonically increasing utility functions. In particular, if $\mathbf{v} \succ \mathbf{v}'$, then $\mathbf{v}$ will be preferred over $\mathbf{v}'$ by all decision-makers. The set of all pairwise Pareto non-dominated vectors is called the Pareto front, denoted $\mathcal{V}^*$. An $\varepsilon$-Pareto front $\mathcal{V}_\varepsilon$ is an approximate Pareto front such that

$\forall \mathbf{v} \in \mathcal{V}^*, \exists \mathbf{v}' \in \mathcal{V}_\varepsilon : \|\mathbf{v} - \mathbf{v}'\|_\infty \leq \varepsilon$. We refer to the least upper bound of the Pareto front as the ideal vector $\mathbf{v}^i$, and the greatest lower bound as the nadir vector $\mathbf{v}^n$.

**Achievement scalarising functions**. Achievement scalarising functions (ASFs) are functions that scalarise a multi-objective problem such that an optimal solution to the single-objective problem is (weakly) Pareto optimal (Miettinen, 1998; Nikulin et al., 2012). Such functions are parameterised by a reference point $\mathbf{r}$, also called the referent, and the points dominating the referent are referred to as the target region. We consider two distinct types of ASFs, known as the order representation case and the order approximation case. Let $s_\mathbf{r}$ be an order representing achievement scalarising function, then for any reference point $\mathbf{r}$, the function $s_\mathbf{r}$ is strictly increasing, i.e. $\mathbf{v} > \mathbf{v}' \implies s_\mathbf{r}(\mathbf{v}) > s_\mathbf{r}(\mathbf{v}')$ and only returns a non-negative value for a vector $\mathbf{v}$ when $\mathbf{v} \succeq \mathbf{r}$. On the other hand, the ASF is order approximating when it is strongly increasing, i.e. $\mathbf{v} \succ \mathbf{v}' \implies s_\mathbf{r}(\mathbf{v}) > s_\mathbf{r}(\mathbf{v}')$ but may give non-negative value to solutions outside the target region. As an example, assume two vectors $\mathbf{v}_1 = (1, 2)$ and $\mathbf{v}_2 = (1, 1)$ such that $\mathbf{v}_1$ Pareto dominates $\mathbf{v}_2$ ($\mathbf{v}_1 \succ \mathbf{v}_2$), but not strictly Pareto dominates it. Hence, for a strictly increasing function $s_\mathbf{r}$, it is possible that $s_\mathbf{r}(\mathbf{v}_1) = s_\mathbf{r}(\mathbf{v}_2)$, while for a strongly increasing function $s_\mathbf{r}$, it is guaranteed that $s_\mathbf{r}(\mathbf{v}_1) > s_\mathbf{r}(\mathbf{v}_2)$. An ASF cannot be strongly increasing while also exclusively attributing non-negative values to vectors that are at least equal to the reference point (Wierzbicki, 1982).

For a set $X$ of feasible solutions, in the order representation case $\mathbf{v}^* = \arg\max_{\mathbf{v} \in X} s_\mathbf{r}(\mathbf{v})$ is guaranteed to be weakly Pareto optimal. Moreover, in the order approximating case $\mathbf{v}^*$ is guaranteed to be Pareto optimal. As such, ASFs come with the advantage that any (weakly) Pareto optimal solution can be obtained by changing the reference point. One example of an ASF that is frequently employed is the augmented Chebyshev scalarisation function (Nikulin et al., 2012; Van Moffaert et al., 2013), which we also utilise in this work.

**Problem setup**. We consider sequential multi-objective decision-making problems, modelled as a multi-objective Markov decision process (MOMDP). An MOMDP is a tuple $M = (\mathcal{S}, \mathcal{A}, T, \mathbf{R}, \gamma)$ where $\mathcal{S}$ is the set of states, $\mathcal{A}$ the set of actions, $T$ the transition function, $\mathbf{R} : \mathcal{S} \times \mathcal{A} \to \mathbb{R}^d$ the vectorial reward function with $d \geq 2$ the number of objectives and $\gamma$ the discount factor. In single-objective RL, it is common to learn a policy that maximises expected return. In an MOMDP, however, there is generally not a single policy that maximises the expected return for all objectives (Hayes et al., 2022). As such, we introduce a partial ordering over policies on the basis of Pareto dominance and say that a policy $\pi \in \Pi$ Pareto dominates another if its expected return, defined $\mathbf{v}^\pi := \mathbb{E}_\pi \left[ \sum_{t=0}^\infty \gamma^t \mathbf{R}(s_t, a_t) \right]$, Pareto dominates the expected return of the other policy.

We aim to learn a Pareto front of memory-based deterministic policies in MOMDPs. Deterministic policies are appropriate for safety-critical settings (Amani et al., 2021), where stochastic policies may have catastrophic outcomes but can Pareto dominate deterministic policies (Hayes et al., 2022). Furthermore, for deterministic policies, it can be shown that memory-based policies may Pareto dominate stationary policies (Roijers & Whiteson, 2017). Formally, we define a deterministic memory-based policy as a Mealy machine $\pi = \langle Q, \pi_\alpha, \pi_\mu, q_I \rangle$ where $Q$ is a set of memory states, $\pi_\alpha : \mathcal{S} \times Q \to \mathcal{A}$ a deterministic next action function, $\pi_\mu : \mathcal{S} \times Q \times \mathcal{A} \times \mathcal{S} \to Q$ the memory update function and $q_I$ the initial memory state. In this setting, it is known that the Pareto front may be non-convex and thus cannot be fully recovered by methods based on linear scalarisation.

## 3 PARETO ORACLE

We introduce a novel concept, called a Pareto oracle, which allows us to obtain Pareto optimal policies in a specified target region. Central to the design of IPRO, presented in Section 4, Pareto oracles supplement a partial Pareto front by discovering new optimal solutions in unexplored regions when possible. We relate Pareto oracles to achievement scalarising functions (Wierzbicki, 1982) and demonstrate their design for our problem setting. Formal proofs of our theoretical results are presented in Appendix A.

### 3.1 DEFINITION AND RELATION TO ASFs

We formally define two variants of a Pareto oracle. These oracles allow us to obtain non-dominated policies in some target region, determined by a reference point $\mathbf{r}$, with the difference between the

variants defining the quality of the returned policy as well as their adherence to the target region. We emphasise that our definitions are tailored to reinforcement learning scenarios, where the feasible solution set consists of policies denoted as $\Pi$. However, these definitions can be readily adapted to apply more broadly across any feasible solution set $X$.

We first introduce weak Pareto oracles which are guaranteed to return a weakly Pareto optimal solution that complies exactly with the target region specified by a reference point.

**Definition 3.1** (Weak Pareto Oracle). A weak Pareto Oracle $\mathcal{O} : \mathbb{R}^d \to \Pi$ maps a vector $\mathbf{r}$ to a weakly Pareto optimal policy $\pi$, such that $\mathbf{v}^\pi \succeq \mathbf{r}$ when such a policy exists. Otherwise, an arbitrary feasible policy $\pi'$ is returned.

The concept of a weak Pareto oracle is closely related to that of an order representing achievement scalarising function (see Section 2). In particular, we can frame the evaluation of an oracle $\mathcal{O}$ with referent $\mathbf{r}$ as the optimisation of an ASF over a set of allowed policies $\Pi$.

**Theorem 3.1.** *Let $s_\mathbf{r}$ be an order representing achievement scalarising function. Then $\mathcal{O}(\mathbf{r}) = \arg\max_{\pi \in \Pi} s_\mathbf{r}(\mathbf{v}^\pi)$ is a valid weak Pareto oracle.*

While this result ensures that weakly optimal solutions can be obtained by proposing referents to an order representing ASF, practical considerations may lead us to favour an order-approximating ASF, which yields Pareto optimal solutions instead. The following definition introduces the concept of an approximate Pareto oracle, for which we subsequently demonstrate that order-approximating ASFs may be used.

**Definition 3.2** (Approximate Pareto Oracle). An approximate Pareto Oracle $\mathcal{O}^\varepsilon : \mathbb{R}^d \to \Pi$ with error $\varepsilon > 0$ maps a vector $\mathbf{r}$ to a Pareto optimal policy $\pi$, such that $\mathbf{v}^\pi \succeq \mathbf{r}$ when a Pareto optimal policy $\pi'$ exists for which $\mathbf{v}^{\pi'} \succeq \mathbf{r} + \varepsilon$. Otherwise, an arbitrary feasible policy $\pi'$ is returned.

We stress that an approximate Pareto oracle is only guaranteed to return a Pareto optimal solution when a solution exists which is at least equal to the referent shifted by some positive value. However, the returned solution itself does not necessarily dominate the shifted referent but is instead guaranteed to be at least equal to the original referent. This relaxation allows for the possibility of obtaining points in the entire target region, rather than only the restricted target region. Finally, in Definitions 3.1 and 3.2 the relation between the expected returns and the referent is non-strict, i.e. $\mathbf{v}^\pi \succeq \mathbf{r}$. When the Pareto oracle instead guarantees that $\mathbf{v}^\pi > \mathbf{r}$, we say the oracle is *boundary-free*.

**Theorem 3.2.** *Let $s_\mathbf{r}$ be an order approximating achievement scalarising function and let $\mathbf{l} \in \mathbb{R}^d$ be a lower bound such that only referents $\mathbf{r}$ are selected when $\mathbf{r} \succeq \mathbf{l}$. Then there is an oracle error $\bar{\varepsilon}$ such that when $\varepsilon \geq \bar{\varepsilon} > 0$, $\mathcal{O}^\varepsilon(\mathbf{r}) = \arg\max_{\pi \in \Pi} s_{\mathbf{r}+\varepsilon}(\mathbf{v}^\pi)$ is a valid approximate Pareto oracle. When $\varepsilon > \bar{\varepsilon}$ the Pareto oracle is boundary-free as well.*

To address the possibility of an order-approximating ASF obtaining a maximum outside the target region, it is necessary to shift the referent upwards by (at least) a problem-specific $\bar{\varepsilon}$. This $\bar{\varepsilon}$ is dependent on an approximation parameter in the ASF, which governs the inclusion of additional points. One potential drawback of Theorem 3.2 then lies in the difficulty of establishing this lower bound. A more practical alternative is to utilise an order-approximating ASF while still optimising for $\arg\max_{\mathbf{v}^\pi \in \Pi} s_\mathbf{r}(\mathbf{v}^\pi)$, as is the case in the weak Pareto oracle. As we demonstrate in Section 5, this strategy proves effective in learning a Pareto front.

To illustrate the difference between a weak and approximate Pareto oracle, we show a possible evaluation of both oracles with a specific referent in Fig. 1. We emphasise that one can devise Pareto oracles that operate independently of ASFs and therefore treat them as a black box in our theoretical analysis in Section 4. We demonstrate this in Appendix A, where we introduce an alternative approach for implementing approximate Pareto oracles making use of constrained MDPs.

### 3.2 Designing a Pareto Oracle

To design both weak and approximate Pareto oracles, we utilise the well-known augmented Chebyshev scalarisation function defined below (Nikulin et al., 2012).

$$s_\mathbf{r}(\mathbf{v}) = \min_{j \in \{1,\dots,d\}} \lambda_j(\mathbf{v}_j - \mathbf{r}_j) + \rho \sum_{j=1}^{d} \lambda_j(\mathbf{v}_j - \mathbf{r}_j) \tag{1}$$

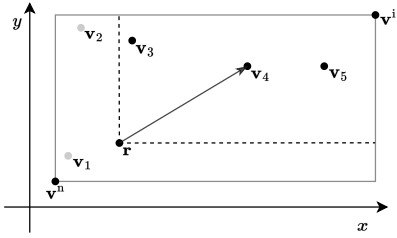 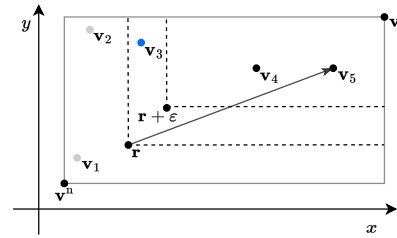

(a) A weak Pareto oracle.                          (b) An approximate Pareto oracle.

Figure 1: A bi-objective problem where a referent $\mathbf{r}$ is presented to a Pareto oracle. Solutions inside the target region are in black, while solutions outside the target region are grey. (a) The weak Pareto oracle returns $\mathbf{v}_4$, which is in the target region but is only weakly Pareto optimal as it is still dominated by $\mathbf{v}_5$. (b) The approximate Pareto oracle returns a Pareto optimal solution $\mathbf{v}_5$, but may fail to find $\mathbf{v}_3$, highlighted in blue.

In this function, $\boldsymbol{\lambda} > 0$ is a weight vector for the different objectives and $\rho$ is an additional parameter which determines the strength of the augmentation term. We set $\boldsymbol{\lambda} = (\mathbf{v}^{\mathrm{i}} - \mathbf{r})^{-1}$ which normalises the improvement of a vector $\mathbf{v}$ relative to the referent $\mathbf{r}$ by the theoretical maximum improvement at the ideal $\mathbf{v}^{\mathrm{i}}$. The purpose of this normalisation is to maintain a balanced scale across all objectives, preventing the dominance of one objective over another.

An important property of this function is that it is concave, which can be verified by noting that a pointwise minimum of affine functions is concave and that adding the augmentation term preserves concavity. Furthermore, it may serve as a boundary-free weak Pareto oracle, or approximate Pareto oracle, depending on the augmentation parameter $\rho$. Intuitively, when $\rho = 0$ the augmentation term is cancelled and the minimum ensures that only vectors in the target region have non-negative values. However, optimising a minimum may result in only weakly Pareto optimal solutions (e.g. $(1, 2)$ and $(1, 1)$ share the same minimum). For $\rho > 0$, the optimal solution will be Pareto optimal (the sum of $(1, 2)$ is greater than that of $(1, 1)$) but may exceed the target region. Due to its versatility, we use Eq. (1) as the ASF in our experimental evaluation.

### 3.2.1   DETERMINISTIC MEMORY-BASED POLICIES

When deterministic policies are necessary for safety or interpretability reasons, it is known that memory-based policies may outperform stationary ones (Roijers & Whiteson, 2017). Intuitively, this is due to the optimal next action not only depending on the current state but also on what has occurred previously. Such memory-based policies are not only relevant in MOMDPs, but also in other related models such as partially observable MDPs.

A common encoding of the memory in MORL is to use the accrued reward at timestep $t$ defined as $\mathcal{R}_t^- = \sum_{k=0}^{t-1} \gamma^k \mathbf{r}_k$ (Hayes et al., 2022). As such, the internal memory update accumulates the observed (discounted) reward, while the policy selects an action based on the current state and accrued reward. In our practical implementation, the memory is added to the observation at each timestep. Since for this policy class, there are no established theoretical results similar to those available for stochastic stationary policies, we propose three modifications of single-objective RL algorithms that have been shown to be effective in practice.

**DQN**. We extend the GGF-DQN algorithm, which optimises for the generalised Gini welfare of the expected returns (Siddique et al., 2020), to optimise any given scalarisation function $f$. We note that GGF-DQN is itself an extension of DQN (Mnih et al., 2015). Concretely, we train a Q-network such that $\mathbf{Q}(s_t, a_t) = \mathbf{r} + \gamma \mathbf{Q}(s_{t+1}, a^*)$ where the optimal action $a^*$ is computed using both the accrued reward as well as the scalarisation function,

$$a^* = \arg\max_{a \in \mathcal{A}} f\left(\mathcal{R}_{t+1}^- + \gamma \mathbf{Q}\left(s_{t+1}, a\right)\right). \tag{2}$$

**Policy gradient**. We modify two well-known policy gradient algorithms, A2C (Mnih et al., 2016) and PPO (Schulman et al., 2017), to optimise $J(\pi) = f(\mathbf{v}^\pi)$, where $f$ is a scalarisation function and

$\pi$ a parameterised policy with parameters $\theta$. For differentiable $f$, we can derive the policy gradient as follows (Reymond et al., 2023),

$$\nabla_\theta J(\pi) = f'(\mathbf{v}^\pi) \cdot \nabla_\theta \mathbf{v}^\pi(s_0). \tag{3}$$

This gradient update computes the regular single-objective gradient for all objectives and takes the dot product with the gradient of the scalarisation function with respect to the expected returns.

Rather than learning a policy with the vanilla policy gradient, we propose to use either the A2C or PPO update per objective instead. For A2C, we use generalised advantage estimation (Schulman et al., 2018) as a baseline, while for PPO we substitute the policy gradient with a clipped surrogate objective function. To ensure that the resulting policy is deterministic, we take actions according to $\arg\max_{a \in \mathcal{A}} \pi(a|s)$ during policy evaluation. Although this potentially leads to significant changes in the policy, effectively employing a policy that differs from the one initially learned, empirical observations suggest that these algorithms typically converge toward deterministic policies in practice.

## 4 ITERATED PARETO REFERENT OPTIMISATION

We introduce Iterated Pareto Referent Optimisation (IPRO) to provably learn a Pareto front in MOMDPs through a decomposition-based approach. IPRO iteratively queries a Pareto oracle and uses the returned solution to gradually reduce the search space. We prove an upper bound on the maximum distance between the solutions found by IPRO and the true Pareto front and guarantee its convergence to a Pareto front. Detailed pseudocode for IPRO can be found in Algorithm 1. We highlight that while IPRO is introduced within the reinforcement learning context, it does not impose specific assumptions about the underlying problem. In fact, it only requires a problem-specific Pareto oracle, indicating its potential for broader application.

---

**Algorithm 1** The IPRO algorithm.

---

**Input:** A boundary-free Pareto oracle $\mathcal{O}$ and a tolerance $\tau$
**Output:** A $\tau$-Pareto front $\mathcal{V}$
 1: Estimate the nadir $\mathbf{v}^n$ and ideal $\mathbf{v}^i$ to establish a bounding box $B$
 2: $\mathcal{V} \leftarrow \{\mathbf{v}_1, \cdots, \mathbf{v}_d\}$
 3: $C \leftarrow \emptyset$
 4: $D^- \leftarrow \{\mathbf{v} \in B \mid \exists \mathbf{v}' \in \mathcal{V}, \mathbf{v}' \succeq \mathbf{v}\}$             ▷ Dominated set (see Appendix B.1)
 5: $D^+ \leftarrow \{\mathbf{v} \in B \mid \exists \mathbf{v}' \in \mathcal{V} \cup C, \mathbf{v} \succeq \mathbf{v}'\}$               ▷ Dominating set
 6: Define $L$ and $U$ to track lower and upper bounds of the Pareto front (see Appendix B.1)
 7: **while** $\max_{\mathbf{u} \in U} \min_{\mathbf{v}' \in \mathcal{V}} \|\mathbf{u} - \mathbf{v}'\|_\infty > \tau$ **do**
 8:     $\mathbf{l} \leftarrow \arg\max_{\mathbf{l} \in L} h(\mathbf{l})$         ▷ Heuristic $h$ for referent selection (see Appendix B.2)
 9:     $\mathbf{v} \leftarrow \mathcal{O}(\mathbf{l})$
10:     **if** $\mathbf{v} > \mathbf{l}$ **then**
11:         $\mathcal{V} \leftarrow \mathcal{V} \cup \{\mathbf{v}\}$
12:     **else**
13:         $C \leftarrow C \cup \{\mathbf{l}\}$
14:     Update $D^-, D^+, L$ and $U$                 ▷ Described in Appendix B.1

---

### 4.1 OVERVIEW

The execution of IPRO unfolds in two main phases, each serving a specific purpose in the algorithm's operation. We provide an overview of these phases below.

**Initialisation and bounding**. In the initial phase, it is necessary to bound the space in which Pareto non-dominated solutions may exist. By definition of the nadir $\mathbf{v}^n$ and ideal $\mathbf{v}^i$ vectors, the box $B$ defined by $\prod_{j=1}^d [\mathbf{v}_j^n, \mathbf{v}_j^i]$ contains all such points, possibly with Pareto optimal vectors on its boundary. Obtaining the ideal (resp. nadir) is possible by maximising (resp. minimising) each objective independently, effectively reducing the MOMDP to a regular MDP that can be solved with conventional methods. Additionally, we subtract a small positive value from the nadir to ensure all potentially Pareto optimal solutions strictly dominate it, implying that they can be retrieved by a boundary-free Pareto oracle.

**Main loop**. In the main loop of IPRO, we iteratively select reference points to present to the Pareto oracle and use the return value to trim sections from the search space. This process continues until the maximum approximation error falls below the user-provided threshold. We determine this maximum approximation error by computing the maximum distance from the points in the upper set $U$ to their closest point on the learned Pareto front $\mathcal{V}$. Informally, the upper set $U$ contains the inner corners of the dominating set $D^+$ that dominates the learned Pareto front. By the definition of a Pareto oracle, we know that the interior of $D^+$ is guaranteed to be infeasible, therefore making the corners on the boundary of $D^+$ the maximal points which may still exist.

In each iteration, a reference point $\mathbf{l}$ is selected from the set of lower points $L$. Analogous to the upper set $U$, the lower set $L$ contains the inner corners of the space dominated by the current Pareto front, denoted as $D^-$. Moreover, every undiscovered point on the Pareto front must strictly dominate at least one lower point, thus implying that we may use these points as referents for the Pareto oracle. If the Pareto oracle finds a solution strictly dominating the referent $\mathbf{l}$, this solution is added to the Pareto front. Otherwise, the referent is added to the set of completed points $C$. Finally, the dominated and dominating sets as well as the lower and upper sets are updated accordingly.

We illustrate the execution of IPRO with different resulting vectors from the Pareto oracle in Fig. 2. While in this illustration all unexplored sections are contained in isolated rectangles, this is a special property when $d = 2$. Moreover, for this special case we can make simplifications, which we detail in Section 4.3. When $d > 2$, however, this property does not hold and therefore necessitates more careful updates of the lower and upper sets and potentially costly referent selection. For an extended discussion of IPRO we refer to Appendix B.

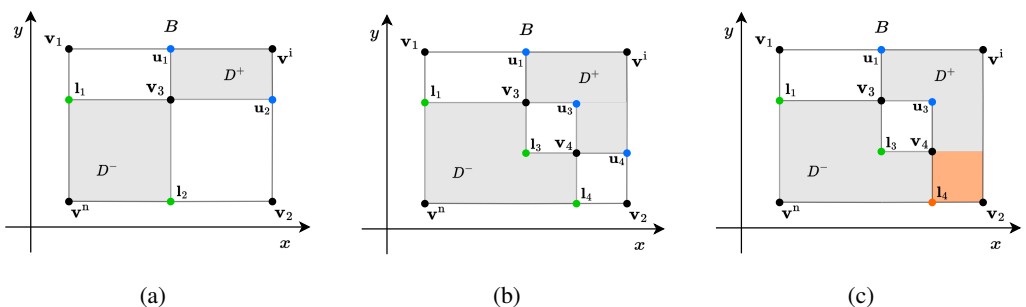

(a)                              (b)                              (c)

Figure 2: (a) The bounding box $B$, delineated with solid lines, is defined by the nadir $\mathbf{v}^{\mathrm{n}}$ and ideal $\mathbf{v}^{\mathrm{i}}$ and contains all Pareto optimal solutions. At timestep $t$, the dominated set $D_t^-$ and dominating set $D_t^+$ are defined by the approximate Pareto front $\mathcal{V}_t = \{\mathbf{v}_1, \mathbf{v}_2, \mathbf{v}_3\}$ and are shaded. The points in the lower set $L$ are highlighted in green, while the points in the upper set $U$ are highlighted in blue. (b) After querying the Pareto oracle with $\mathbf{l}_2$, $\mathbf{v}_4$ is added to the Pareto front and the set of lower points $L$ and upper points $U$ are updated to represent the new corners of $D^-$ and $D^+$ respectively. (c) When a Pareto oracle cannot find a feasible solution strictly dominating $\mathbf{l}_4$, it is added to the completed set $C$ and the shaded orange area is added to $D^+$.

## 4.2 Convergence and Guarantees

The core idea behind IPRO revolves around the systematic reduction of remaining volume within the bounding box, achieved through iterative queries to a Pareto oracle. Here, we present theoretical results proving that IPRO converges to an (approximate) Pareto front, as well as demonstrate an upper bound on the maximum approximation error at every timestep. Formal proofs for our results can be found in Appendix C.

**Upper bounding the error**. Let the true approximation error $\varepsilon_t^*$ to the Pareto front $\mathcal{V}^*$ at timestep $t$ be defined as $\max_{\mathbf{v}^* \in \mathcal{V}^* \setminus \mathcal{V}_t} \min_{\mathbf{v} \in \mathcal{V}_t} \| \mathbf{v}^* - \mathbf{v} \|_\infty$. Since the upper set $U$ contains the maximal points which may still be in the Pareto front, we can search for the point having the largest $l_\infty$ distance from the current Pareto front $\mathcal{V}_t$, resulting in an upper bound on the true approximation error. Note that when using an approximate Pareto oracle $\mathcal{O}^\varepsilon$ and user-provided tolerance $\tau$, we assume that $\tau \geq \varepsilon$ as the overall approximation error is lower bounded by the oracle error $\varepsilon$.

**Theorem 4.1.** *Let $\mathcal{V}^*$ be the true Pareto front, $\mathcal{V}_t$ the approximate Pareto front obtained by IPRO and $\varepsilon_t^*$ the true approximation error at timestep $t$. Then the following inequality holds,*

$$\max_{\mathbf{u} \in U_t} \min_{\mathbf{v} \in \mathcal{V}_t} \|\mathbf{u} - \mathbf{v}\|_\infty \geq \varepsilon_t^*. \tag{4}$$

One can verify this result in Fig. 2b where $U = \{\mathbf{u}_1, \mathbf{u}_3, \mathbf{u}_4\}$ contains the maximal points which may still be in the Pareto front. Note that while $\varepsilon$-Pareto fronts are commonly computed with regards to the $l_\infty$ norm, the guarantee can be extended more generally to any $p$-norm.

**Convergence to a Pareto front**. As IPRO progresses, the sequence of errors generated by Theorem 4.1 can be shown to be monotonically decreasing and converges to zero. Intuitively, this can be observed in Fig. 2b where the retrieval of a new Pareto optimal point reduces the distance to the points in the upper set. Additionally, the closure of a section, illustrated in Fig. 2c, results in the removal of the upper point which subsequently reduces the remaining search space. Since IPRO terminates when the true approximation error is guaranteed to be at most equal to the tolerance $\tau$, this results in a $\tau$-Pareto front.

**Theorem 4.2.** *Given a boundary-free weak (resp. approximate) Pareto oracle and tolerance $\tau \geq 0$ (resp. $\tau > 0$), IPRO converges to a $\tau$-Pareto front.*

While the result concerning weak Pareto oracles might initially appear stronger due to the possibility of setting the tolerance to zero, it is important to note that such oracles guarantee only weakly Pareto optimal solutions for each query, potentially making them less appealing in practice. Finally, we contribute a guarantee that IPRO finishes in a finite number of iterations when the tolerance is strictly positive, which is especially useful in practical applications.

**Corollary 4.2.1.** *Given a tolerance $\tau > 0$, IPRO finishes in a finite number of iterations using a boundary-free weak or approximate Pareto oracle.*

### 4.3 IPRO-2D: A Specialised Version for Bi-Objective Problems

While IPRO is applicable to problems with $d \geq 2$ objectives, updating the lower and upper sets as well as selecting a new referent may be costly. Therefore, we introduce a dedicated variant, for bi-objective problems ($d = 2$), IPRO-2D, where substantial simplifications are possible.

As shown in Fig. 2, all remaining sections within the bounding box $B$ manifest as isolated rectangles, with each lower and upper point precisely defining one such rectangle. When a new Pareto optimal solution is found, updating the lower and upper sets can be done by adding at most two new points for both sets, each on one side of the adjusted boundary. Moreover, calculating the volume of each rectangle is straightforward, making it possible to construct a priority queue that prioritises the processing of larger rectangles to ensure a rapid decrease in the upper bound of the error. The maximum error can be computed by taking the rectangle with the maximum distance between its lower and upper point, rather than performing the full max-min operation in Eq. (4).

As a final adjustment, we modify the weighting for the augmented Chebyshev scalarisation function used in the Pareto oracle. Rather than weighting a vector $\mathbf{v}$ by the distance between the referent $\mathbf{l} \in L$, and the ideal $\mathbf{v}^i$, we normalise the ASF according to the distance between the lower and upper points that make up the isolated rectangle.

## 5 Experiments

We assess IPRO and IPRO-2D on three standard benchmark environments (Alegre et al., 2022) and employ the modified versions of DQN, A2C, and PPO proposed in Section 3.2.1 as Pareto oracles to optimise the augmented Chebyshev scalarisation function in Eq. (1). Experiments are repeated across five seeds. We record the hypervolume in each iteration and estimate the search space coverage by combining the volumes from the dominated and dominating sets and dividing by the bounding box volume. All experimental details can be found in Appendix D and code will be made publicly available upon acceptance.

**Deep Sea Treasure** ($d = 2$). Deep Sea Treasure (DST) is a benchmark with deterministic dynamics and a discrete state space where a submarine seeks treasure while minimising fuel consumption. In

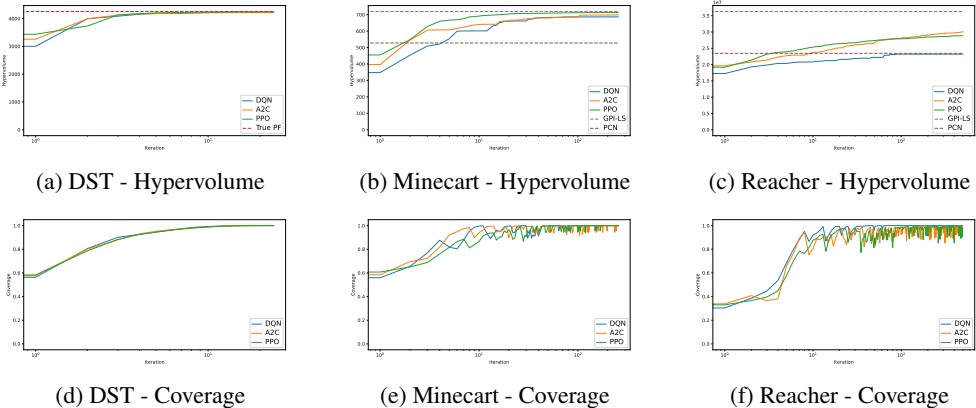

Figure 3: The hypervolume and coverage results for IPRO. The shaded area around each curve indicates the 95-percentile interval. Note that the x-axis is on a log scale.

DST, the true Pareto front is known and includes optimal solutions in concave regions (Vamplew et al., 2011), making it impossible for convex hull algorithms to recover all Pareto optimal solutions. When pairing IPRO-2D with any of the three Pareto oracles, the obtained hypervolume closely approximates that of the true Pareto front (Fig. 3a) and its coverage (Fig. 3d) rapidly approaches one, indicating a complete overview of the search space.

**Minecart** ($d = 3$). Minecart is a stochastic environment with a continuous state space where the agent collects two ore types while minimising fuel consumption (Abels et al., 2019). Since the true Pareto front is unknown, we calculate the mean hypervolume using the five best seeds for two baseline methods, PCN and GPI-LS, from the MORL-Baselines project (Felten et al., 2023). PCN learns deterministic Pareto optimal policies (Reymond et al., 2022), while GPI-LS learns a convex hull (Alegre et al., 2023). We find that IPRO achieves a higher hypervolume than PCN and is competitive with GPI-LS with all Pareto oracles. This outcome is particularly promising given that GPI-LS leverages the knowledge of the convex nature of the Pareto front in the Minecart environment, which is not a necessary assumption in IPRO. We highlight that in Fig. 3e, coverage may decrease in certain iterations. In theory, the sequence of coverages is monotonically increasing. However, due to a backtracking method added on top of IPRO to improve robustness, coverage may temporarily drop in practice (see Appendix B.3).

**MO-Reacher** ($d = 4$). MO-Reacher is a deterministic environment featuring a continuous state space with four balls arranged in a circle, with the goal of minimising the distance to each ball. Here, the true Pareto front is unknown, only leaving the baselines for comparison. We find that IPRO is competitive to PCN, even surpassing it when paired with the policy gradient algorithms, but a performance gap remains when compared to GPI-LS. It is worth noting that GPI-LS employs iterative fine-tuning of prior policies, while IPRO constructs a new Pareto optimal policy from scratch in each iteration. We hypothesise that a similar addition can be made to IPRO and consider this a promising direction for future work.

## 6 CONCLUSION

We introduce IPRO, an algorithm that provably learns a Pareto front in MOMDPs by iteratively proposing referents to a Pareto oracle and using the returned solution to trim sections from the search space. We formally define such Pareto oracles and present their implementation for different policy classes. We show that IPRO converges to a Pareto front and comes with strong guarantees with respect to the approximation error. Our empirical analysis of IPRO finds that it converges close to the true Pareto front and does so for a variety of Pareto oracles. For future work, we plan to apply IPRO to problems beyond MORL and explore alternative Pareto oracle implementations.

## 7 REPRODUCIBILITY STATEMENT

For the theoretical results presented in this work, we offer formal proofs, accompanying definitions, and necessary assumptions within their respective sections of the appendix. Specifically, the results pertaining to Pareto oracles are presented in Appendix A, while the results for IPRO are elaborated upon in Appendix C. To facilitate the practical implementation of IPRO, we provide an expanded discussion in Appendix B covering both the pseudocode of the theoretical algorithm as well as further improvements to ensure robustness. Moreover, we share an anonymised implementation of IPRO, IPRO-2D and all of our proposed Pareto oracles, and commit to releasing the final code publicly upon acceptance. Lastly, a detailed account of the experimental evaluation of IPRO is provided in Appendix D which also encompasses all hyperparameter values used to reproduce the results when using our implementations of the algorithms.

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

# A    THEORETICAL RESULTS FOR PARETO ORACLES

We present formal proofs for the theoretical results in Section 3. These results develop the concept of a Pareto oracle and relate it to achievement scalarising functions. While we utilise Pareto oracles as a subroutine in IPRO to provably obtain a Pareto front, they may also be of independent interest in other settings. As an additional contribution, we relate Pareto oracles to constrained MDPs, thus demonstrating their applicability beyond the ASFs considered in this work.

## A.1    DESIGNING A PARETO ORACLE

In Section 3.1 we defined Pareto oracles and subsequently related them to achievement scalarising functions. Here, we provide formal proof of the established connections. To establish the notation, let $X$ be the set of feasible solutions and define a mapping $f : X \to \mathbb{R}^d$ which maps a solution to its $d$-dimensional return. Let us further define the Euclidean distance function between a point $\mathbf{v} \in \mathbb{R}^d$ and a set $S \subseteq \mathbb{R}^d$ as $\mathrm{dist}(\mathbf{v}, S) = \inf_{\mathbf{s} \in S} \|\mathbf{v} - \mathbf{s}\|$. Finally, let $\mathbb{R}_\delta^d = \{\mathbf{v} \in \mathbb{R}^d \mid \mathrm{dist}(\mathbf{v}, \mathbb{R}_{\geq 0}^d) \leq \delta \|\mathbf{v}\|\}$, where $\delta$ is a fixed scalar in $[0, 1)$. Using this notation, we define both order representing and order approximating ASFs following the formalisation by Miettinen (1998).

**Definition A.1.** We say an ASF $s_{\mathbf{r}} : \mathbb{R}^d \to \mathbb{R}$ is order representing when $\forall \mathbf{r} \in \mathbb{R}^d, \forall x, y \in X$ with $f(x) = \mathbf{x}$ and $f(y) = \mathbf{y}$, $s_{\mathbf{r}}$ is strictly increasing such that $\mathbf{x} > \mathbf{y} \implies s_{\mathbf{r}}(\mathbf{x}) > s_{\mathbf{r}}(\mathbf{y})$. In addition, $s_{\mathbf{r}}(\mathbf{r}) = 0$ and

$$\{\mathbf{v} \in \mathbb{R}^d \mid s_{\mathbf{r}}(\mathbf{v}) \geq 0\} = \mathbf{r} + \mathbb{R}_{\geq 0}^d. \tag{5}$$

**Definition A.2.** We say an ASF $s_{\mathbf{r}} : \mathbb{R}^d \to \mathbb{R}$ is order approximating when $\forall \mathbf{r} \in \mathbb{R}^d, \forall x, y \in X$ with $f(x) = \mathbf{x}$ and $f(y) = \mathbf{y}$, $s_{\mathbf{r}}$ is strongly increasing such that $\mathbf{x} \succ \mathbf{y} \implies s_{\mathbf{r}}(\mathbf{x}) > s_{\mathbf{r}}(\mathbf{y})$. In addition, $s_{\mathbf{r}}(\mathbf{r}) = 0$ and with $\delta > \bar{\delta} \geq 0$

$$\mathbf{r} + \mathbb{R}_{\bar{\delta}}^d \subset \{\mathbf{v} \in \mathbb{R}^d \mid s_{\mathbf{r}}(\mathbf{v}) \geq 0\} \subset \mathbf{r} + \mathbb{R}_\delta^d. \tag{6}$$

These definitions can be applied to the reinforcement learning setting where the set of feasible solutions is a policy class $\Pi$ and the quality of a policy $\pi \in \Pi$ is determined by its expected return $\mathbf{v}^\pi$. Using these definitions, we provide a formal proof for Theorem 3.1 which we first restate below.

**Theorem 3.1.** *Let $s_{\mathbf{r}}$ be an order representing achievement scalarising function. Then $\mathcal{O}(\mathbf{r}) = \arg\max_{\pi \in \Pi} s_{\mathbf{r}}(\mathbf{v}^\pi)$ is a valid weak Pareto oracle.*

*Proof.* Let $s_{\mathbf{r}}$ be an order representing achievement scalarising function and define a Pareto oracle $\mathcal{O} : \mathbb{R}^d \to \Pi$ such that, $\mathcal{O}(\mathbf{r}) = \arg\max_{\pi \in \Pi} s_{\mathbf{r}}(\mathbf{v}^\pi) = \pi^*$. Denote the expected return of $\pi^*$ as $\mathbf{v}^*$. We first consider the case when $\mathbf{v}^* \not\succeq \mathbf{r}$. By Eq. (5) this implies that $s_{\mathbf{r}}(\mathbf{v}^*) < 0$. This guarantees that no feasible weakly Pareto optimal policy $\pi'$ exists with expected return $\mathbf{v}$' such that $\mathbf{v}' \succeq \mathbf{r}$, as otherwise $s_{\mathbf{r}}(\mathbf{v}') \geq 0 > s_{\mathbf{r}}(\mathbf{v}^*)$ and thus $\pi^*$ would not have been returned as the maximum.

We now consider the case when $\mathbf{v}^* \succeq \mathbf{r}$. Then $\pi^*$ is guaranteed to be weakly Pareto optimal. By contradiction, if $\pi^*$ is not weakly Pareto optimal, another policy $\pi'$ exists such that $\mathbf{v}' > \mathbf{v}^*$. However, this would imply that $s_{\mathbf{r}}(\mathbf{v}') > s_{\mathbf{r}}(\mathbf{v}^*)$ and thus $\pi^*$ would not have been returned as the maximum. $\qquad\square$

We provide a similar result using order-approximating ASFs instead. While such ASFs enable the Pareto oracle to return Pareto optimal solutions rather than only weakly optimal solutions, the quality of the oracle with respect to the target region becomes dependent on the approximation parameter $\delta$ of the ASF. The core idea in the proof of Theorem 3.2 is that we can define a lower bound on the shift necessary to ensure only feasible solutions in the target region have a non-negative value. When feasible solutions exist in the shifted target region, we can then conclude by the strongly increasing property of the ASF that the maximum is Pareto optimal.

**Theorem 3.2.** *Let $s_{\mathbf{r}}$ be an order approximating achievement scalarising function and let $\mathbf{l} \in \mathbb{R}^d$ be a lower bound such that only referents $\mathbf{r}$ are selected when $\mathbf{r} \succeq \mathbf{l}$. Then there is an oracle error $\bar{\varepsilon}$ such that when $\varepsilon \geq \bar{\varepsilon} > 0$, $\mathcal{O}^\varepsilon(\mathbf{r}) = \arg\max_{\pi \in \Pi} s_{\mathbf{r}+\varepsilon}(\mathbf{v}^\pi)$ is a valid approximate Pareto oracle. When $\varepsilon > \bar{\varepsilon}$ the Pareto oracle is boundary-free as well.*

*Proof.* Let $\mathbf{l}$ be the lower bound for all referents $\mathbf{r}$. We define $\bar{\varepsilon}$ to be the minimal shift such that all feasible solutions with non-negative values for an order-approximating ASF $s_{1+\bar{\varepsilon}}$ with the shifted referent $\mathbf{l} + \bar{\varepsilon}$ are inside the box $B(\mathbf{l}, \mathbf{v}^{\mathbf{i}})$ defined by the lower bound and ideal. The lower bound on $\bar{\varepsilon}$ is clearly zero which implies that no shift is necessary. We now define an upper bound for this shift which ensures that no feasible solution has a non-negative value except potentially $\mathbf{l}$ itself.

Recall the definition of $\mathbb{R}^d_{\delta} = \{ \mathbf{v} \in \mathbb{R}^d \mid \mathrm{dist}(\mathbf{v}, \mathbb{R}^d_{\geq 0}) \leq \delta \|\mathbf{v}\| \}$, where $\delta$ is a fixed scalar in $[0, 1)$. We refer to $\mathbf{l} + \mathbb{R}^d_{\delta}$ as the extended target region. Suppose there exists a point in this extended target region $\mathbf{v} \in \mathbf{l} + \mathbb{R}^d_{\delta}$ such that $\mathbf{l} \succ \mathbf{v}$. This implies we can write $\mathbf{v} = \mathbf{l} + \mathbf{x}$, where $\mathbf{x}$ is a non-positive vector. However, this then further implies that, $\mathrm{dist}(\mathbf{x}, \mathbb{R}^d_{\geq 0}) = \inf_{\mathbf{s} \in \mathbb{R}^d_{\geq 0}} \|\mathbf{x} - \mathbf{s}\| = \|\mathbf{x}\|$ as 0 is the closest point in $\mathbb{R}^d_{\geq 0}$ for a non-positive vector. However, for $\delta \in [0, 1)$ it cannot be true that $\|\mathbf{x}\| \leq \delta \|\mathbf{x}\|$. Therefore, there exists no point in $\mathbf{l} + \mathbb{R}^d_{\delta}$ that is dominated by $\mathbf{l}$. As such, for all points $\mathbf{v}$ in the extended target region that are not equal to $\mathbf{l}$, there must be a dimension $j \in \{1, \ldots, d\}$ such that $\mathbf{v}_j > \mathbf{l}_j$. Consider now the shift imposed by the $l_{\infty}$ distance between the lower point $\mathbf{l}$ and ideal $\mathbf{v}^{\mathbf{i}}$. This ensures that all points in the extended target region except $\mathbf{l}$ are strictly above the ideal in at least one dimension, further implying that they are infeasible by the definition of the ideal. As such, $\|\mathbf{v}^{\mathbf{i}} - \mathbf{l}\|_{\infty}$ is an upper bound for $\bar{\varepsilon}$.

Let us now formally define $\bar{\varepsilon}$ for an order approximating ASF with approximation constant $\delta$,

$$\bar{\varepsilon} = \inf \left\{ 0 < \varepsilon \leq \|\mathbf{v}^{\mathbf{i}} - \mathbf{l}\|_{\infty} \mid \left( \mathbf{l} + \varepsilon + \mathbb{R}^d_{\delta} \right) \cap \{ \mathbf{v} \in \mathbb{R}^d \mid \mathbf{v}^{\mathbf{i}} \succeq \mathbf{v} \} \subseteq B(\mathbf{l}, \mathbf{v}^{\mathbf{i}}) \right\}. \tag{7}$$

In Fig. 4 we illustrate that this shift ensures all feasible solutions with non-negative values are inside the box. Observe, however, that by the nature of this shift, it can also ensure that some feasible solutions in the bounding box are excluded from the non-negative set.

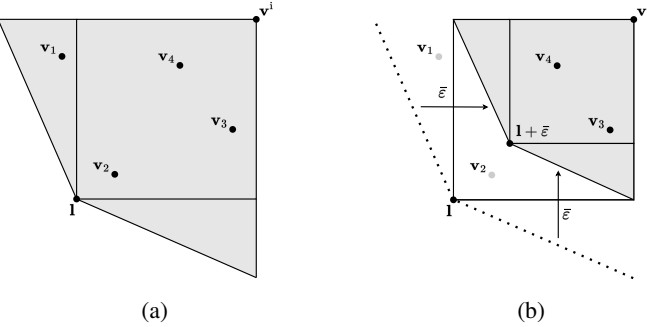

|       |       |
|:-----:|:-----:|
| (a)   | (b)   |

Figure 4: (a) A possible non-negative set (shaded) for an order-approximating ASF with referent $\mathbf{l}$. (b) Shifting $\mathbf{l}$ by $\bar{\varepsilon}$ ensures that all feasible solutions with non-negative values are in the box $B(\mathbf{l}, \mathbf{v}^{\mathbf{i}})$.

Let us now show that the Pareto oracle $\mathcal{O}^{\varepsilon}(\mathbf{r}) = \arg\max_{\pi \in \Pi} s_{\mathbf{r}+\varepsilon}(\mathbf{v}^{\pi})$ with $\varepsilon \geq \bar{\varepsilon}$ functions as required for the referent $\mathbf{l}$. Assume there exists a Pareto optimal optimal $\pi'$ with expected return $\mathbf{v}'$ such that $\mathbf{v}' \succeq \mathbf{l} + \varepsilon$. Then $s_{\mathbf{l}+\varepsilon}(\mathbf{v}') \geq 0$ and therefore the maximisation will return a non-negative solution $\pi^*$ with expected returns $\mathbf{v}^*$. By the definition of $\bar{\varepsilon}$ we know that all feasible solutions $\pi$ with non-negative value $s_{\mathbf{l}+\varepsilon}(\mathbf{v}^{\pi})$ satisfy the condition $\mathbf{v}^{\pi} \succeq \mathbf{l}$ and therefore $\mathbf{v}^* \succeq \mathbf{l}$. Moreover, as the ASF is guaranteed to be strongly increasing, there exists no policy $\pi$ such that $\mathbf{v}^{\pi} \succ \mathbf{v}^*$ and therefore $\pi^*$ is Pareto optimal.

Given the lower bound $\mathbf{l}$, for all referents $\mathbf{r}$ such that $\mathbf{r} \succeq \mathbf{l}$ and with $\varepsilon \geq \bar{\varepsilon}$, the Pareto oracle remains valid. To see this, observe that $\mathbf{r} = \mathbf{l} + \mathbf{x}$ where $\mathbf{x}$ is now a non-negative vector. Then,

$$\begin{aligned} & \left( \mathbf{l} + \varepsilon + \mathbb{R}^d_{\delta} \right) \cap \{ \mathbf{v} \in \mathbb{R}^d \mid \mathbf{v}^{\mathbf{i}} \succeq \mathbf{v} \} \subseteq B(\mathbf{l}, \mathbf{v}^{\mathbf{i}}) \\ \implies & \left( \mathbf{l} + \varepsilon + \mathbb{R}^d_{\delta} \right) \cap \{ \mathbf{v} \in \mathbb{R}^d \mid \mathbf{v}^{\mathbf{i}} - \mathbf{x} \succeq \mathbf{v} \} \subseteq B(\mathbf{l}, \mathbf{v}^{\mathbf{i}} - \mathbf{x}). \end{aligned} \tag{8}$$

This implication can be shown by contradiction. Assume that,

$$\exists \mathbf{v} \in \left( \mathbf{l} + \varepsilon + \mathbb{R}^d_{\delta} \right) \cap \{ \mathbf{v} \in \mathbb{R}^d \mid \mathbf{v}^{\mathbf{i}} - \mathbf{x} \succeq \mathbf{v} \} \text{ and } \mathbf{v} \notin B(\mathbf{l}, \mathbf{v}^{\mathbf{i}} - \mathbf{x}). \tag{9}$$

However, by definition of $\mathbf{v}$, $\mathbf{v}^{\mathrm{i}} - \mathbf{x} \succeq \mathbf{v}$ and

$$\mathbf{v} \in \left(\mathbf{l} + \varepsilon + \mathbb{R}_\delta^d\right) \cap \{\mathbf{v} \in \mathbb{R}^d \mid \mathbf{v}^{\mathrm{i}} - \mathbf{x} \succeq \mathbf{v}\}$$
$$\implies \mathbf{v} \in \left(\mathbf{l} + \varepsilon + \mathbb{R}_\delta^d\right) \cap \{\mathbf{v} \in \mathbb{R}^d \mid \mathbf{v}^{\mathrm{i}} \succeq \mathbf{v}\}$$
$$\implies \mathbf{v} \in B(\mathbf{l}, \mathbf{v}^{\mathrm{i}})$$
$$\implies \mathbf{v} \succeq \mathbf{l}.$$

As $\mathbf{v} \succeq \mathbf{l}$ and $\mathbf{v}^{\mathrm{i}} - \mathbf{x} \succeq \mathbf{v}$ this implies $\mathbf{v} \in B(\mathbf{l}, \mathbf{v}^{\mathrm{i}} - \mathbf{x})$, which is a contradiction. Therefore $\left(\mathbf{l} + \varepsilon + \mathbb{R}_\delta^d\right) \cap \{\mathbf{v} \in \mathbb{R}^d \mid \mathbf{v}^{\mathrm{i}} - \mathbf{x} \succeq \mathbf{v}\} \subseteq B(\mathbf{l}, \mathbf{v}^{\mathrm{i}} - \mathbf{x})$. By a rigid transformation and recalling that $\mathbf{r} = \mathbf{l} + \mathbf{x}$, we obtain,

$$\left(\mathbf{r} + \varepsilon + \mathbb{R}_\delta^d\right) \cap \{\mathbf{v} \in \mathbb{R}^d \mid \mathbf{v}^{\mathrm{i}} \succeq \mathbf{v}\} \subseteq B(\mathbf{r}, \mathbf{v}^{\mathrm{i}}). \tag{10}$$

We can subsequently apply the same reasoning to establish the validity of the Pareto oracle for the lower bound $\mathbf{l}$ to all dominating referents $\mathbf{r}$.

Finally, for $\varepsilon > \bar{\varepsilon}$ the approximate Pareto oracle becomes boundary-free as a consequence of $\bar{\varepsilon}$ being the infimum ensuring that all feasible solutions have non-negative values. $\qquad\square$

## A.2 Designing a Pareto Oracle

We formalise the design of Pareto oracles. Firstly, we show that the augmented Chebyshev scalarisation function in Eq. (1) can be used as a weak and approximate Pareto oracle. In addition, we demonstrate that this scalarisation function can also be used to derive theoretically sound Pareto oracles for the class of stochastic stationary policies. Finally, we demonstrate that Pareto oracles need not be designed using an ASF and contribute an alternative approach using constrained MDPs.

### A.2.1 Augmented Chebyshev Scalarisation Function

We show that the augmented Chebyshev scalarisation function we employ in our work is a boundary-free weak Pareto oracle or approximate Pareto oracle depending on the augmentation parameter $\rho$. The boundary-free property in particular is useful as this allows us to "complete" sections of the search space in IPRO when the Pareto oracle does not retrieve a solution in the target region. For clarity, we first restate the scalarisation function below.

$$s_{\mathbf{r}}(\mathbf{v}) = \min_{j \in \{1,\dots,d\}} \lambda_j (\mathbf{v}_j - \mathbf{r}_j) + \rho \sum_{j=1}^d \lambda_j (\mathbf{v}_j - \mathbf{r}_j)$$

**Corollary A.0.1.** *Equation* (1) *can be used to construct a valid boundary-free weak Pareto oracle when $\rho = 0$ and an approximate Pareto oracle when $\rho > 0$.*

*Proof.* When $\rho = 0$, Eq. (1) reduces to $s_{\mathbf{r}}(\mathbf{v}) = \min_{j \in \{1,\dots,d\}} \lambda_j (\mathbf{v}_j - \mathbf{r}_j)$, which is a well-known order-represeting ASF (Miettinen, 1998). Moreover, when $\mathbf{v}$ is on the boundary, $s_{\mathbf{r}}(\mathbf{v}) = 0$ as there exists an index $j \in \{1, \dots, d\}$ for which $\mathbf{v}_j = \mathbf{r}_j$. This guarantees that $\mathbf{v} > \mathbf{r}$ implies $s_{\mathbf{r}}(\mathbf{v}) > 0$ and therefore the weak Pareto oracle is boundary-free as well. Finally, when $\rho > 0$, it is again a well-known fact that the resulting ASF is order-approximating (Miettinen, 1998). $\qquad\square$

### A.2.2 Stochastic Stationary Policies

To design a Pareto oracle for stochastic stationary policies $\pi : \mathcal{S} \to \Delta(\mathcal{A})$, we consider the augmented Chebyshev scalarisation function in Eq. (1) again. Rather than maximising $s_{\mathbf{r}}$ directly, we can instead compute an optimal state-action occupancy measure $d_\pi$ induced by the policy $\pi$ over a set of admissible state-action occupancies $\mathcal{K}$ (Zahavy et al., 2021). By the concavity of $s_{\mathbf{r}}$ this results in a convex MDP, shown below.

$$\arg\max_{d_\pi \in \mathcal{K}} s_{\mathbf{r}} \left( \sum_{s,a} \mathbf{r}(s,a) d_\pi(s,a) \right) \tag{11}$$

The benefit of this reformulation is that it can be solved using a variety of techniques that come with strong theoretical guarantees. For instance, Zhang et al. (2020) propose a policy gradient method that

converges to the global optimum. Moreover, Zahavy et al. (2021) introduce a meta-algorithm using standard RL algorithms that converges to the optimal solution with any tolerance, given reasonably low-regret algorithms. Finally, it can be shown that, for any convex MDP, a mean-field game can be constructed such that a Nash equilibrium in the game is an optimum in the convex MDP (Geist et al., 2022). While we do not focus on this policy class in our experimental evaluation, these results indicate the potential for designing a Pareto oracle tailored to stochastic stationary policies.

### A.2.3 Constrained MDPs

To accompany our results relating ASFs and Pareto oracles, we provide an additional contribution which sheds light on the versatility of these oracles. Concretely, we demonstrate that a Pareto oracle can be implemented using constrained MDPs which may again be solved with conventional techniques. This further validates the usage of abstract Pareto oracles in the theoretical results for IPRO, rather than relying on achievement scalarising functions directly.

A constrained Markov decision process (Achiam et al., 2017) is an MDP, augmented with a set $C$ of $m$ auxiliary cost functions $C_j : \mathcal{S} \times \mathcal{A} \times \mathcal{S} \to \mathbb{R}$ and related limit $c_j$. Let $J_{C_j}(\pi)$ denote the expected discounted return of policy $\pi$ for the auxiliary cost function $C_j$. The set of feasible policies from a given class of policies $\Pi$ is then $\Pi_C = \{\pi \in \Pi \mid \forall i, J_{C_j}(\pi) \geq c_j\}$. Finally, the reinforcement learning problem in a CMDP is as follows,

$$\pi^* = \arg\max_{\pi \in \Pi_C} v^\pi. \tag{12}$$

We demonstrate that an approximate Pareto oracle can be implemented using a constrained MDP, where the constraints ensure that the target region is respected and the scalar reward function is designed such that only Pareto optimal policies are returned as the optimal solution. Moreover, as constrained MDPs consider only policies in the shifted target region, the resulting Pareto oracle is immediately boundary-free with respect to the original target region.

**Theorem A.1.** *Let $M = (\mathcal{S}, \mathcal{A}, T, \mathbf{R}, \gamma)$ be an MOMDP with $d$ objectives. For a given oracle error $\varepsilon > 0$ and referent $\mathbf{r}$, we define a constrained MDP with the same states, actions, transition function and discount factor as $M$. Furthermore, the set of auxiliary cost functions corresponds to the original reward vector $\mathbf{R}$ with limits $\mathbf{r} + \varepsilon$ and the scalar reward function is the sum of the original reward vector. Then $\mathcal{O}^\varepsilon(\mathbf{r}) = \arg\max_{\pi \in \Pi_C} v^\pi$ is a valid boundary-free approximate Pareto oracle.*

*Proof.* Assume the construction outlined in the theorem and that there exists a Pareto optimal policy $\pi$ such that $\mathbf{v}^\pi \succeq \mathbf{r} + \varepsilon$. Then $\Pi_C$ is non-empty and the Pareto oracle $\mathcal{O}^\varepsilon(\mathbf{r}) = \arg\max_{\pi \in \Pi_C} v^\pi$ returns a Pareto optimal policy $\pi^*$ with expected return $\mathbf{v}^*$ such that $\mathbf{v}^* \succ \mathbf{r}$. If $\pi^*$ is not Pareto optimal, there exists a policy $\pi'$ with expected return $\mathbf{v}'$ such that $\mathbf{v}' \succ \mathbf{v}^*$. This then implies that,

$$\sum_{j \in \{1,...,d\}} v_j' > \sum_{j \in \{1,...,d\}} v_j^* \tag{13}$$

which leads to a contradiction. Furthermore, by definition of a constrained MDP the solution is in the target region and boundary-free for $\varepsilon > 0$. $\qquad \square$

## B Additional Discussion of IPRO

We offer a more in-depth analysis of our algorithm, Iterated Pareto Referent Optimisation (IPRO). This extended discussion includes additional pseudocode for IPRO and a breakdown of its implementation. Additionally, we delve into the topic of referent selection throughout the execution of IPRO and examine alterations to the theoretical algorithm that ensure IPRO's practical applicability when paired with imprecise Pareto oracles.

### B.1 Implementation of IPRO

IPRO follows an inner-outer loop structure. During its execution, IPRO tracks the current Pareto front and excluded sections and iteratively proposes referents to a Pareto oracle. Finally, the obtained Pareto front is pruned to eliminate dominated solutions. We show detailed pseudocode in

Algorithm 1. To simplify the notation, we denote the input to IPRO as an oracle $\mathcal{O}$ and tolerance $\tau$ but note that it may also be an approximate Pareto oracle $\mathcal{O}^\varepsilon$ with $\tau \geq \varepsilon$ and assume $\mathcal{O}$ returns the multi-objective returns of the solution directly, rather than the solution itself. Following Theorem 3.2, we further assume the oracle error $\varepsilon$ to be greater than the lower bound error $\bar{\varepsilon}$ when using an order approximating ASF and where this lower bound is computed with respect to the nadir $\mathbf{v}^n$.

**Tracking the dominated and dominating set**. In the context of IPRO, it is essential to keep track of the dominated set $D^-$ and dominating set $D^+$. These sets encompass points excluded from further consideration, either because they are dominated by a point on the Pareto front or because they strictly dominate a point on the front. In the latter case, the definition of both a weak and approximate Pareto oracle guarantees that no feasible policy exists to attain this result; otherwise, it would have been returned instead of the point it strictly dominates.

Rather than explicitly monitoring these sets, we maintain the points on the Pareto front $\mathcal{V}$ and the completed set $C$. From these sets, we can compute the hypervolume covered by $D^-$ and $D^+$ which provides an estimate of the overall coverage achieved by IPRO.

**Tracking the lower and upper set**. An important concept in IPRO is the notion of the lower and upper set. Informally, the lower set contains a strict lower bound for all Pareto optimal points not yet discovered, while the upper set contains a (possibly non-strict) upper bound on these remaining solutions. Visually, the lower set contains the inner corners of the dominated set, while the upper set contains the inner corners of the dominating set. We formally demonstrate in Appendix C that this indeed results in the required lower and upper bounds.

The lower and upper sets play a crucial role in deriving both our convergence guarantee and runtime guarantee on the largest distance to remaining Pareto optimal solutions. Consequently, tracking these sets is of paramount importance. After the initialisation phase, the lower set encompasses the nadir, while the upper set contains the ideal. We now explain in detail how the lower set is tracked and note that a similar update can be defined for the upper set.

After initialisation, upon introducing a new point $\mathbf{v}^*$ to the Pareto front, we examine the lower set to identify points that have become strictly dominated by it. If $\mathbf{l}$ is such a dominated point, it is replicated $d$ times, each time adjusting one dimension of the vector to align with the boundary of the newly added point $\mathbf{v}^*$. We note that this approach may generate points within the dominated set $D^-$. Therefore, we perform a final pruning step to eliminate points not on the boundary. Pseudocode for updating the lower set is provided in Algorithm 2.

---

**Algorithm 2** Computing the lower set.

---

**Input:** A bounding box $B$, previous lower set $L_{t-1}$ and new point $\mathbf{v}^*$
**Output:** A lower set $L_t$
1:   $L_t \leftarrow \{\}$
2: **for** $\mathbf{l} \in L_{t-1}$ **do**
3:      **if** $\mathbf{v}^* > \mathbf{l}$ **then**
4:          **for** $j \in [d]$ **do**
5:              $\mathbf{l}' \leftarrow \mathbf{l}$
6:              $\mathbf{l}'_j \leftarrow \mathbf{v}^*_j$
7:              $L_t \leftarrow L_t \cup \{\mathbf{l}'\}$
8:      **else**
9:          $L_t \leftarrow L_t \cup \{\mathbf{l}\}$
10: $L_t \leftarrow \text{PRUNE}(L_t)$

---

**Postprocessing**. While IPRO inherently requires no postprocessing, there exists a possibility that weakly Pareto optimal points added during IPRO's execution may have become dominated by subsequent additions to the Pareto front. While this does not impact the hypervolume of the final Pareto front, it could pose a challenge for decision-makers in selecting their preferred solution. To streamline the set presented to decision-makers, we enhance the obtained Pareto front by eliminating dominated points. In our practical implementation, we further include all solutions rejected by the Pareto oracle in the approximate Pareto front before the pruning step. While theoretically redundant with an exact Pareto oracle, this step may, in practice, reveal additional Pareto optimal solutions.

## B.2 Referent Selection

The iterative process constructed in IPRO involves proposing referents to a Pareto oracle. Naturally, a crucial question arises: how should these referents be chosen? While our theoretical outcomes are not contingent on a particular method for selecting referents, we propose the use of the hypervolume improvement heuristic. This heuristic suggests referents that, when incorporated into the dominating set, would yield the greatest increase in hypervolume. Intuitively, referents with a high hypervolume improvement indicate a large unexplored region that dominates it, suggesting that new Pareto optimal solutions may be found there. We define this formally in Definition B.1

**Definition B.1** (Hypervolume Improvement). Let $\mathbf{r} \in \mathbb{R}^d$ be a reference point and $HV(S, \mathbf{r})$ be the hypervolume of a set $S$ with respect to the reference point. The hypervolume improvement $HVI$ of a point $\mathbf{v} \in \mathbb{R}^d$ is defined as the contribution of $\mathbf{v}$ to the hypervolume when added to $S$, i.e.

$$HVI(\mathbf{v}, S, \mathbf{r}) = HV(S \cup \{\mathbf{v}\}, \mathbf{r}) - HV(S, \mathbf{r}) \tag{14}$$

It is worth noting that computing the hypervolume improvement for a large number of points can become prohibitively expensive, as it necessitates a new hypervolume computation for each point. This computational cost is one of the main reasons why the dedicated 2D variant of IPRO is more efficient. Concretely, in the two-dimensional case, all remaining area is made up of isolated rectangles described by one lower and upper point. For these rectangles, we can efficiently compute the area and keep a priority queue of rectangles with the greatest remaining area. Selecting the next referent can then be done by taking the first rectangle from the priority queue and using its lower point as the referent.

Finally, we highlight that depending on prior knowledge regarding the Pareto front's shape or specific regions of interest, the referent selection method can be readily adapted and alternative metrics for assessing improvement such as sparsity, could be incorporated into the process.

## B.3 Robustness through Backtracking

As discussed in Section 5, we introduce a modification to IPRO to accommodate imperfect Pareto oracles. Specifically, we integrate a backtracking procedure into IPRO, which is triggered whenever a solution is returned that violates a decision made in a previous iteration.

To implement this robustness feature, we maintain a sequence denoted as $\{(\mathbf{r}_t, \mathbf{v}_t)\}_{t \in \mathbb{N}}$ where each pair records the reference point and the retrieved vector for a given iteration. When, at time step $t + 1$, the returned vector $\mathbf{v}_{t+1}$ strictly dominates a point $\mathbf{c} \in C_t$ or $\mathbf{v}^* \in \mathcal{V}_t$, it indicates an incorrect oracle evaluation in a previous iteration. To address this issue, we initiate a replay of the sequence of pairs. Let $\bar{t}$ denote the time step at which the incorrect result was initially returned. For all time steps $n \in \{0, \ldots, \bar{t} - 1\}$, we replay the pairs using the standard IPRO updates and consider $\mathbf{v}_{t+1}$ as the retrieved solution for $\mathbf{r}_{\bar{t}}$.

Subsequently, for all time steps $n \in \{\bar{t}, \ldots, t\}$, we apply different conditions than in the original iterations and leverage the transitivity property of Pareto dominance to recycle the pairs. Specifically, if $\mathbf{v}_n > \mathbf{r}_n$, it guarantees that $\mathbf{v}_n$ is (weakly) Pareto optimal. This information is used to check whether there exists a point $\mathbf{l} \in L$ that is strictly dominated by $\mathbf{v}_n$. If such a point is identified, $(\mathbf{l}, \mathbf{v}_n)$ can be added using the standard IPRO update rule.

Finally, when $\mathbf{v}_n$ fails to strictly dominate the referent $\mathbf{r}_n$, it implies that $\mathbf{r}_n$ would have been included in the completed set. In this scenario, we verify whether there exists a lower point that Pareto dominates $\mathbf{r}_n$. If such a lower point is found, we add $\mathbf{l}$ to the completed set instead.

## C Theoretical Results for IPRO

In this section, we provide the omitted proofs for IPRO from Section 4. These results establish both the upper bound to the true approximation error as well as guarantee convergence to the true Pareto front in the limit or an approximate Pareto front in a finite number of iterations.

### C.1 Definitions and Assumptions

Before presenting the proofs for IPRO, it is necessary to formally define the sets that are tracked in IPRO. Let $B = \prod_{j=1}^{d} [\mathbf{v}_j^n, \mathbf{v}_j^i]$ be the bounding box defined by a strict lower bound of the true nadir $\mathbf{v}^n$ and the ideal $\mathbf{v}^i$. The set $\mathcal{V}_t$ contains the obtained Pareto front at timestep $t$ while the completed set $C_t$ contains the referents for which the Pareto oracle failed to find a strictly dominating solution. We then define the dominated and dominating set $D^-$ and $D^+$ as follows.

**Definition C.1.** The dominated set $D_t^-$ at timestep $t$ contains all points in the bounding box that are dominated by or equal to a point in the current Pareto front, i.e.

$$D_t^- = \{\mathbf{v} \in B \mid \exists \mathbf{v}' \in \mathcal{V}_t, \mathbf{v}' \succeq \mathbf{v}\}. \tag{15}$$

**Definition C.2.** The dominating set $D_t^+$ at timestep $t$ contains all points in the bounding box that dominate or are equal to a point in the union of the current Pareto front and completed referents, i.e.

$$D_t^+ = \{\mathbf{v} \in B \mid \exists \mathbf{v}' \in \mathcal{V}_t \cup C_t, \mathbf{v} \succeq \mathbf{v}'\}. \tag{16}$$

Note that in the definition for the dominating set, we consider not only those points dominated by the current Pareto front but also the points dominated by the referents that failed to result in new solutions.

During the execution of IPRO, it is necessary to recognise the remaining unexplored sections. For this, we define the reachable boundaries of the dominated and dominating set which together delineate the remaining search space. Let $\overline{S}$ be the closure of a subset $S$ in some topological space and $\partial S$ be its boundary. By a slight abuse of notation, we say that $\partial D_t^- = \overline{(B \setminus D_t^-)} \cap \overline{D_t^-}$ is the boundary of $D_t^-$ in $B$, which is itself a subset of Euclidean space. Defining the boundary of $D_t^+$ analogously, we define the reachable boundaries as follows.

**Definition C.3.** The reachable boundary of $D_t^-$, denoted $\partial^r D_t^-$ at timestep $t$ is defined as,

$$\partial^r D_t^- = \partial D_t^- \setminus D_t^+. \tag{17}$$

**Definition C.4.** The reachable boundary of $D_t^+$, denoted $\partial^r D_t^+$ at timestep $t$ is defined as,

$$\partial^r D_t^+ = \partial D_t^+ \setminus D_t^-. \tag{18}$$

In addition, the interior of $D_t^-$ is then defined as $\text{int } D_t^- = D_t^- \setminus \partial D_t^-$ and the interior of $D_t^+$ analogously. We illustrate these subsets in Fig. 5.

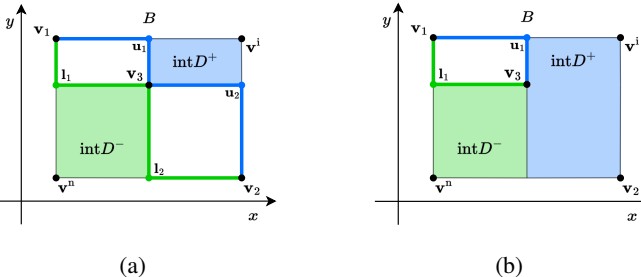

|          (a)          |          (b)          |

Figure 5: (a) The reachable boundaries of $D_t^-$ (green) and $D_t^+$ (blue) indicated with solid lines and their interiors (shaded) when no section is completed. (b) When completing the section at $\mathbf{l}_2$, parts of the reachable boundary at timestep $t$ become unreachable at timestep $t+1$.

For the reachable boundaries of the dominated and dominating set, we define two important subsets, namely the lower and upper sets. The lower set contains the points on the reachable boundary of the dominated set $D^-$ such that no other point on the reachable boundary exists which is dominated by it. Similarly, the upper set contains the points on the reachable boundary of $D^+$ such that no other point exists on the reachable boundary that dominates it. Conceptually, these points are the inner corners of their respective boundary as can be observed in Fig. 5.

**Definition C.5** (Lower Set). The lower set at timestep $t$ is defined as,

$$L_t = \left\{ \mathbf{l} \in \partial^r D_t^- \mid \nexists \mathbf{v} \in \partial^r D_t^-, \mathbf{l} \succ \mathbf{v} \right\}. \tag{19}$$

**Definition C.6** (Upper Set). The upper set at timestep $t$ is defined as,

$$U_t = \left\{ \mathbf{u} \in \partial^r D_t^+ \mid \nexists \mathbf{v} \in \partial^r D_t^+, \mathbf{v} \succ \mathbf{u} \right\}. \tag{20}$$

Before continuing with our proofs, we make two explicit assumptions. First, to ensure that IPRO still functions as intended when utilising a boundary-free approximate Pareto oracle, we assume that the user-provided tolerance is at least as high as the oracle tolerance.

**Assumption C.1.** For a given approximate Pareto oracle $\mathcal{O}^\varepsilon$ with error $\varepsilon$ and user-provided tolerance $\tau$, we assume that $\tau \geq \varepsilon$.

Finally, we assume that the problem is not trivial and there exist unexplored regions in the bounding box $B$ after finding the first $d$ weakly Pareto optimal solutions. This assumption is not hindering, as we can easily verify whether out of the first $d$ weakly Pareto optimal solutions, there is a single Pareto optimal one and terminate IPRO if this is the case. Note that this assumption utilises a pruning algorithm PPRUNE which takes as input a set of points and returns only the Pareto optimal points (Roijers & Whiteson, 2017).

**Assumption C.2.** For the given MOMDP $M$ with $d$-dimensional reward function, we assume that for the initial Pareto front $\mathcal{V}_0 = \{\mathbf{v}_1, \ldots, \mathbf{v}_d\}$ it is guaranteed that $|\text{PPRUNE}(\mathcal{V}_0)| > 1$.

## C.2 SUPPORTING LEMMAS

We provide supporting lemmas that formalise the contents of the sets defined in Appendix C.1 and their relation to the remaining feasible solutions. Concretely, we first demonstrate that the interior of the dominating set contains only infeasible points, which is a consequence of having a strictly positive distance to the boundary. Combined with the dominating set, which inherently contains only dominated solutions, we can then significantly reduce the search space that is left to explore.

**Lemma C.1.** *During IPRO's execution, it is guaranteed that all points in the interior of the dominating set are infeasible.*

*Proof.* Recall that the interior of the dominating set is defined as follows,

$$\text{int } D_t^+ = D_t^+ \setminus \partial D_t^+. \tag{21}$$

Let $\mathbf{v} \in \text{int } D_t^+$ be a point in the interior of the dominating set. Then there exists an open ball centred around $\mathbf{v}$ with a strictly positive radius $r$ such that $B_r(\mathbf{v}) \subseteq \text{int } D_t^+$. Let $\mathbf{v}' \in B_r(\mathbf{v})$ be a point in the ball such that $\mathbf{v} > \mathbf{v}'$ which can be obtained by taking $\mathbf{v}$ and subtracting a value $\delta \in (0, r)$. Since $\mathbf{v}' \in \text{int } D_t^+$, the definition of the dominating set (Definition C.2) ensures that there exists a point $\bar{\mathbf{v}} \in \mathcal{V}_t \cup C_t$ such that $\mathbf{v}' \succeq \bar{\mathbf{v}}$. By the transitivity of Pareto dominance, we then have that $\mathbf{v} > \bar{\mathbf{v}}$.

Let us now consider the two cases for $\bar{\mathbf{v}}$. Assume first that $\bar{\mathbf{v}} \in \mathcal{V}_t$. If $\mathbf{v}$ is a feasible solution and knowing that $\mathbf{v} > \bar{\mathbf{v}}$ implies that $\bar{\mathbf{v}}$ is not weakly Pareto optimal. Therefore, $\bar{\mathbf{v}}$ would not have been returned by a weak or approximate Pareto oracle. As such, $\mathbf{v}$ must be infeasible.

Finally, when $\bar{\mathbf{v}} \in C_t$ it was added after a Pareto oracle could not find a feasible solution strictly dominating the lower point $\mathbf{l} = \bar{\mathbf{v}}$ implying again that $\mathbf{v}$ is infeasible. $\qquad \square$

Given the result for the infeasible solutions, we now focus instead on the remaining feasible solutions. Here, we demonstrate that all feasible solutions are strictly lower bounded by the lower set and upper bounded by the upper set.

**Lemma C.2.** *During IPRO's execution, the lower set contains a strict lower bound for all remaining feasible solutions, i.e.,*

$$\mathbf{v} \in B \setminus (\text{int } D_t^+ \cup D_t^-) \implies \exists \mathbf{l} \in L_t, \mathbf{v} > \mathbf{l}. \tag{22}$$

*Proof.* Let $\mathbf{v}$ be a remaining feasible solution. Then it cannot be in the dominated set, as this implies it is dominated by a point on the current Pareto front, nor can it be in the interior of $D_t^+$ as this was guaranteed to be infeasible following Lemma C.1. However, $\mathbf{v}$ can still be on the reachable boundary of the dominating set when using a weak Pareto oracle. As such, we may indeed write in Eq. (22) that $\mathbf{v} \in B \setminus (\text{int } D_t^+ \cup D_t^-)$.

Recall that in IPRO, the nadir $\mathbf{v}^{\text{n}}$ of the bounding box $B$ is initialised to a guaranteed strict lower bound of the true nadir (see line two of Algorithm 1). Therefore, for all $\mathbf{v} \in B \setminus (\text{int } D_t^+ \cup D_t^-)$ we can connect a strictly decreasing line between $\mathbf{v}$ and $\mathbf{v}^{\text{n}}$. Moreover, either $\mathbf{v}^{\text{n}} \in \partial D_t^-$ or this line must intersect $\partial D_t^-$ at some point $\bar{\mathbf{v}}$ for which it is subsequently guaranteed that $\mathbf{v} > \bar{\mathbf{v}}$.

Let $\mathbf{v} \in B \setminus (\text{int } D_t^+ \cup D_t^-)$ be a feasible solution and $\bar{\mathbf{v}} \in \partial D_t^-$ be a point on the boundary of $D_t^-$ such that $\mathbf{v} > \bar{\mathbf{v}}$. Suppose, however, that $\bar{\mathbf{v}}$ is not on the reachable boundary. Then, the definition of the reachable boundary implies that $\bar{\mathbf{v}} \in D_t^+$ (see Definition C.3). However, as $\mathbf{v} > \bar{\mathbf{v}}$ this implies that $\mathbf{v}$ is in the interior of $D_t^+$ which was guaranteed to be infeasible by Lemma C.2. Therefore, $\bar{\mathbf{v}}$ must be on the reachable boundary of $D_t^-$. By definition of the lower set, this further implies there exists a lower point $\mathbf{l} \in L_t$ for which $\bar{\mathbf{v}} \succeq \mathbf{l}$, finally guaranteeing that $\mathbf{v} > \mathbf{l}$. $\qquad\square$

We provide an analogous result for the upper set where we demonstrate that it contains an upper bound for all remaining feasible solutions.

**Lemma C.3.** *During IPRO's execution, the upper set contains an upper bound for all remaining feasible solutions, i.e.,*

$$\mathbf{v} \in B \setminus (\text{int } D_t^+ \cup D_t^-) \implies \exists \mathbf{u} \in U_t, \mathbf{u} \succeq \mathbf{v}. \tag{23}$$

*Proof.* As the ideal $\mathbf{v}^{\text{i}}$ is initialised to the true ideal, we may apply the same proof as for Lemma C.2 using Pareto dominance rather than strict Pareto dominance. In contrast to Lemma C.2 however, $\mathbf{v}$ may be on the reachable boundary of the dominating set $\partial^r D_t^+$. In this case, the definition of the upper set guarantees the existence of an upper point $\mathbf{u} \in U_t, \mathbf{u} \succeq \mathbf{v}$. $\qquad\square$

## C.3 PROOF OF THEOREM 4.1

We now prove Theorem 4.1 which guarantees an upper bound on the true approximation error at any timestep. In fact, this upper bound follows almost immediately from the supporting lemmas shown in Appendix C.2. Utilising the fact that the upper points are a guaranteed upper bound for the remaining feasible solutions, we can compute the upper point that maximises the distance to its closest point on the current approximation of the Pareto front. Recall that at timestep $t$ the true approximation error $\varepsilon_t^*$ is defined as $\max_{\mathbf{v}^* \in \mathcal{V}^* \setminus \mathcal{V}_t} \min_{\mathbf{v} \in \mathcal{V}_t} \| \mathbf{v}^* - \mathbf{v} \|_\infty$.

**Theorem 4.1.** *Let $\mathcal{V}^*$ be the true Pareto front, $\mathcal{V}_t$ the approximate Pareto front obtained by IPRO and $\varepsilon_t^*$ the true approximation error at timestep $t$. Then the following inequality holds,*

$$\max_{\mathbf{u} \in U_t} \min_{\mathbf{v} \in \mathcal{V}_t} \| \mathbf{u} - \mathbf{v} \|_\infty \geq \varepsilon_t^*. \tag{24}$$

*Proof.* Observe that all remaining Pareto optimal solutions must be feasible and we can therefore derive from Lemmas C.1 and C.3 that

$$\forall t \in \mathbb{N}, \forall \mathbf{v}^* \in \mathcal{V}^* \setminus \mathcal{V}_t, \exists \mathbf{u} \in U_t : \mathbf{u} \succeq \mathbf{v}^*. \tag{25}$$

From Eq. (25) we can then conclude the following upper bound,

$$\max_{\mathbf{u} \in U_t} \min_{\mathbf{v} \in \mathcal{V}_t} \| \mathbf{u} - \mathbf{v} \|_\infty \geq \max_{\mathbf{v}^* \in \mathcal{V}^* \setminus \mathcal{V}_t} \min_{\mathbf{v} \in \mathcal{V}_t} \| \mathbf{v}^* - \mathbf{v} \|_\infty = \varepsilon_t^*. \tag{26}$$

Note that this holds as the maximum over the upper points is guaranteed to be at least as high as the maximum over all remaining points on the Pareto front. $\qquad\square$

## C.4 PROOF OF THEOREM 4.2 AND COROLLARY 4.2.1

To conclude the theoretical contributions for IPRO, we show that it is guaranteed to converge to a $\tau$-Pareto front where $\tau$ is the user-provided tolerance. Moreover, when using a weak Pareto oracle, the $\tau$ may be set to $0$ and the true Pareto front is obtained in the limit. For practical purposes, however, setting $\tau > 0$ ensures that IPRO converges after a finite number of iterations.

**Theorem 4.2.** *Given a boundary-free weak (resp. approximate) Pareto oracle and tolerance $\tau \geq 0$ (resp. $\tau > 0$), IPRO converges to a $\tau$-Pareto front.*

*Proof.* We show that, when ignoring the tolerance parameter $\tau$, the sequence of errors generated by IPRO is a monotonically decreasing sequence with its infimum at zero. By the monotone convergence theorem, the limit of this sequence is zero. When incorporating the tolerance $\tau$, IPRO stops when the approximation error is guaranteed to be at most $\tau$ therefore resulting in a $\tau$-Pareto front.

Let us first denote,
$$\max_{\mathbf{u} \in U_t} \min_{\mathbf{v} \in \mathcal{V}_t} \|\mathbf{u} - \mathbf{v}\|_\infty = \varepsilon_t \tag{27}$$
and demonstrate that $\forall t \in \mathbb{N} : \varepsilon_t \leq \varepsilon_{t-1}$.

By construction of $\mathcal{V}_t$ it is guaranteed that $\mathcal{V}_{t-1} \subseteq \mathcal{V}_t$. Therefore,
$$\max_{\mathbf{u} \in U_t} \min_{\mathbf{v} \in \mathcal{V}_t} \|\mathbf{u} - \mathbf{v}\|_\infty \leq \max_{\mathbf{u} \in U_t} \min_{\mathbf{v} \in \mathcal{V}_{t-1}} \|\mathbf{u} - \mathbf{v}\|_\infty \tag{28}$$
Considering $U_t$ instead, it was guaranteed that $U_{t-1}$ contains the maximal points which may still be in the Pareto front at timestep $t-1$. Therefore, for all upper points $\mathbf{u} \in U_t$ there must exist an old upper point $\bar{\mathbf{u}} \in U_{t-1}$ such that $\bar{\mathbf{u}} \succeq \mathbf{u}$. As such, we conclude that
$$\max_{\mathbf{u} \in U_t} \min_{\mathbf{v} \in \mathcal{V}_t} \|\mathbf{u} - \mathbf{v}\|_\infty \leq \max_{\mathbf{u} \in U_{t-1}} \min_{\mathbf{v} \in \mathcal{V}_{t-1}} \|\mathbf{u} - \mathbf{v}\|_\infty \tag{29}$$
and thus $\forall t \in \mathbb{N} : \varepsilon_t \leq \varepsilon_{t-1}$.

We now show that the sequence of errors $(\varepsilon_t)_{t \in \mathbb{N}}$ converges to zero. Clearly, 0 is a lower bound for the errors as there is no further approximation error when $U_t$ is the empty set. Then, for any $\beta > 0$, we need to show that $\beta$ is not a lower bound of $(\varepsilon_t)_{t \in \mathbb{N}}$. Suppose there exists such a $\beta$. Then we can write that,
$$\forall \mathbf{v} \in \mathcal{V}_t, \exists \mathbf{u} \in U_t : \max(\mathbf{u} - \mathbf{v}) \geq \beta. \tag{30}$$
For simplicity, assume there is a single $\bar{\mathbf{u}}$ for which Eq. (30) holds. Let $\mathbf{l} \in L_t$ be a lower point such that $\bar{\mathbf{u}} > \mathbf{l}$. Note that this $\mathbf{l}$ must exist by Lemma C.2.

Denote the box spanned by $\mathbf{l}$ and $\bar{\mathbf{u}}$ as $B(\mathbf{l}, \bar{\mathbf{u}})$. Then there is a point $\mathbf{v}$ inside of this box such that $\|\bar{\mathbf{u}} - \mathbf{v}\|_\infty < \beta$. Proposing $\mathbf{l}$ to a Pareto oracle, it is guaranteed that either the box $B(\mathbf{l}, \bar{\mathbf{u}})$ is closed, rendering the maximum distance to zero, or that a new Pareto optimal point $\mathbf{v}^*$ is retrieved. If $\mathbf{v}^* \succeq \mathbf{v}$ the distance is below the threshold. Otherwise, IPRO adds a lower point $\mathbf{l}'$ such that $\mathbf{l}' \succ \mathbf{l}$ and $\mathbf{v} \in B(\mathbf{l}', \bar{\mathbf{u}})$. Repeating this process, either the box is closed for some future lower point or a closer Pareto optimal solution is found which decreases the bound.

Recall that we assume a single $\bar{\mathbf{u}}$ for which Eq. (30) holds. However, if there were multiple $\mathbf{u}$ such that their maximum is greater than $\beta$, we can repeat the same argument. This then implies there is no $\beta > 0$ which is a lower bound for $(\varepsilon_t)_{t \in \mathbb{N}}$ and therefore the sequence of errors must also converge to 0 by the monotone convergence theorem.

Theorem 4.1 guarantees that the true approximation error is upper bounded by $\varepsilon_t$. Since IPRO terminates when this upper bound is at most equal to the tolerance $\tau$, it is guaranteed to converge to a $\tau$-Pareto front. $\qquad\square$

Finally, Theorem 4.2 can be straightforwardly amended to establish IPRO's convergence in a finite number of iterations given a strictly positive tolerance.

**Corollary 4.2.1.** *Given a tolerance $\tau > 0$, IPRO finishes in a finite number of iterations using a boundary-free weak or approximate Pareto oracle.*

*Proof.* This follows immediately from the monotone convergence theorem and Theorem 4.2. Concretely, the decreasing sequence of errors $(\varepsilon_t)_{t \in \mathbb{N}}$ converges to 0 and thus for every $\tau > 0$, there must exist a timestep $T$ such that $\varepsilon_T < \tau$ as otherwise $\tau$ would be a lower bound. $\qquad\square$

# D  EXPERIMENT DETAILS

In this section, we provide details concerning the experimental evaluation presented in Section 5. Concretely, we discuss the selection of baselines and environments and provide the hyperparameters used in our experiments.

### D.1 BASELINES

We first discuss the baselines we employed to evaluate the performance of IPRO.

**GPI-LS**. Generalised Policy Improvement - Linear Support (GPI-LS) is an algorithm designed for obtaining the convex hull of deterministic policies Alegre et al. (2023). As mentioned in Section 2, this solution set is appropriate in scenarios where decision-makers are guaranteed to have linear utility functions. Similar to IPRO, GPI-LS relies on a decomposition approach to reduce the multi-objective problem into efficiently solvable single-objective problems. However, since GPI-LS assumes convexity of the Pareto front, it enables the utilisation of conventional reinforcement learning methods. In contrast, IPRO avoids imposing such constraints on the shape of the Pareto front, necessitating substantial adjustments (see Section 3.2).

GPI-LS is currently the state-of-the-art in various benchmarks. To ensure a fairer comparison between IPRO and GPI-LS, we retain all Pareto optimal policies generated by GPI-LS during its evaluation rather than only the policies in the convex hull. Furthermore, in both Minecart and MO-Reacher, we observe a relatively small difference in hypervolume between the convex hull and the Pareto front. This observation further supports the comparison between GPI-LS and IPRO. Finally, we do not consider GPI-LS in the Deep Sea Treasure environment as the Pareto front is concave and it is thus only able to retrieve the two extremal policies.

**PCN**. Pareto Conditioned Networks (PCN) is a method specifically designed to learn a Pareto front of deterministic policies in MOMDPs. PCN trains a single neural network on a range of desired trade-offs, to generalise over the full set of Pareto optimal policies. This is achieved by learning to predict the "return-to-go" from any state and selecting the action that most closely reaches the returns of the chosen trade-off. While PCN was primarily intended to operate in deterministic environments, it can also be evaluated in stochastic environments. We note that, while MO-Reacher is a deterministic environment, Minecart is not, which may explain its relatively poor performance in this environment.

### D.2 ENVIRONMENTS

To focus solely on IPRO's performance, we initialise each experiment with predefined minimal and maximal points to establish the bounding box of the environment. It is important to emphasise that these points can be obtained using conventional reinforcement learning algorithms without requiring any modifications, justifying their omission from our evaluation process.

**Deep Sea Treasure (DST)**. As the Pareto front is known in DST, we initialise IPRO with $(124, -19)$ and $(0, 0)$ as the maximal points and switch these for the minimal points. We set the discount factor to 1, signifying no discounting, and maintain a fixed time horizon of 50 timesteps for each episode. The hypervolumes shown in Fig. 3 were calculated using $(0, -50)$ as the reference point. We note that we one-hot encode the observations due to the discrete nature of the state space. Finally, a tolerance $\tau$ of 0 was set to allow IPRO to find the complete Pareto front in this environment.

**Minecart**. In the Minecart environment, we set $\gamma = 0.98$ to align with related work. Rather than directly determining the maximal and minimal points from the Pareto front, we provide unattainable points that do not contribute to the hypervolume, ensuring a fair comparison with our baselines. For minimal points, IPRO is initialised with the nadir $(-1, -1, -200)$ for each dimension. For maximal points, we consider the nadir and set each dimension to its theoretical maximum: $(1.5, -1, -200), (-1, 1.5, -200), (-1, -1, 0)$. Our reference point is also the nadir and the time horizon is 1000. A tolerance of $1 \times 10^{-5}$ was used.

**MO-Reacher**. In the Reacher environment, where no Pareto front is known, the bounding box again needs to be estimated. For minimal points, we use $(-50, -50, -50, -50)$ in each dimension, and similarly, set this vector to 40 for each dimension for the maximal points. The discount factor $\gamma$ is set to 0.99. The reference point is again set to the nadir, a time horizon of 50 was used and tolerance was set to $1 \times 10^{-5}$.

### D.3 HYPERPARAMETERS

To conclude this section, in Table 2 we provide a description of all hyperparameters used in our Pareto oracles and the algorithms for which they apply. Finally, in Tables 3 to 5 we give the hyperparameter values used in our reported experiments.

Table 2: A description of the relevant hyperparameters.

| Parameter | Algorithm | Description |
| --- | --- | --- |
| scale | DQN, A2C, PPO | Scale the output of Eq. (1) to generate a greater learning signal |
| $\rho$ | DQN, A2C, PPO | Augmentation parameter from Eq. (1) |
| global_steps | DQN, A2C, PPO | Number of global steps to learn one policy |
| critic_hidden | DQN, A2C, PPO | Number of hidden neurons per layer for the critic |
| lr_critic | DQN, A2C, PPO | The learning rate for the critic |
| actor_hidden | A2C, PPO | Number of hidden neurons per layer for the actor |
| lr_actor | A2C, PPO | Learning rate for the actor |
| n_steps | A2C, PPO | Number of environment interactions before each update |
| gae_lambda | A2C, PPO | $\lambda$ parameter for generalised advantage estimation |
| normalise_advantage | A2C, PPO | Normalise the advantage |
| e_coef | A2C, PPO | Entropy loss coefficient to compute the overall loss |
| v_coef | A2C, PPO | Value loss coefficient to compute the overall loss |
| max_grad_norm | A2C, PPO | Maximum gradient norm |
| clip_coef | PPO | Clip coefficient used in the PPO surrogate objective |
| num_envs | PPO | Number of parallel environments to run in |
| anneal_lr | PPO | Anneal the learning rate over time |
| clip_range_vf | PPO | Clipping range for the value function |
| update_epochs | PPO | Number of update epochs to execute |
| num_minibatches | PPO | Number of minibatches to divide a batch in |
| batch_size | DQN | Batch size for each update |
| buffer_size | DQN | Size of the replay buffer |
| soft_update | DQN | Multiplication factor for the soft update |
| epsilon_start | DQN | Starting exploration probability |
| epsilon_end | DQN | Final exploration probability |
| exploration_frac | DQN | Explore for a fraction of the given total timesteps |
| learning_start | DQN | Only start learning after a number of episodes |

Table 3: The hyperparameters used in the DQN oracles.

| Parameter | DST | Minecart | MO-Reacher |
| --- | --- | --- | --- |
| scale | 100 | 10 | 500 |
| $\rho$ | 0.001 | 0.001 | 0.01 |
| global_steps | 6.0e+04 | 1.0e+06 | 1.0e+05 |
| critic_hidden | (256, 256, 256) | (128, 128) | (256, 256, 256) |
| lr_critic | 0.0003 | 0.0003 | 0.001 |
| batch_size | 32 | 64 | 32 |
| buffer_size | 1.0e+04 | 1.0e+05 | 5.0e+04 |
| soft_update | 0.1 | 0.1 | 0.1 |
| epsilon_start | 0.7 | 0.5 | 0.5 |
| epsilon_end | 0.2 | 0.2 | 0.1 |
| exploration_frac | 0.1 | 0.1 | 0.3 |
| learning_start | 1.0e+04 | 2.0e+03 | 1.0e+03 |

Table 4: The hyperparameters used in the A2C oracles.

| Parameter | DST | Minecart | MO-Reacher |
|---|---|---|---|
| scale | 100 | 100 | 1 |
| $\rho$ | 0 | 0.001 | 0.01 |
| global_steps | 2.0e+05 | 1.5e+06 | 5.0e+05 |
| critic_hidden | (128, 128, 128) | (128,) | (128, 128, 128) |
| lr_critic | 0.0005 | 0.001 | 0.0007 |
| actor_hidden | (128, 128, 128) | (128,) | (128, 128, 128) |
| lr_actor | 0.0001 | 0.001 | 0.0007 |
| n_steps | 16 | 32 | 128 |
| gae_lambda | 0.95 | 1 | 1 |
| normalise_advantage | False | False | True |
| e_coef | 0.1 | 0.1 | 0.01 |
| v_coef | 0.5 | 0.5 | 0.1 |
| max_grad_norm | 10 | 50 | 1 |

Table 5: The hyperparameters used in the PPO oracles.

| Parameter | DST | Minecart | MO-Reacher |
|---|---|---|---|
| scale | 500 | 500 | 1 |
| $\rho$ | 0.005 | 0.01 | 0.01 |
| global_steps | 5.0e+05 | 1.5e+06 | 7.5e+05 |
| critic_hidden | (32, 32, 32) | (128, 128) | (64,) |
| lr_critic | 0.0005 | 0.0001 | 0.001 |
| actor_hidden | (64, 64) | (128, 128) | (64,) |
| lr_actor | 0.0005 | 0.001 | 0.001 |
| n_steps | 128 | 128 | 64 |
| gae_lambda | 0.95 | 1 | 0.95 |
| normalise_advantage | False | True | False |
| e_coef | 0.05 | 0.01 | 0.1 |
| v_coef | 0.3 | 0.1 | 0.1 |
| max_grad_norm | 1 | 0.5 | 0.5 |
| clip_coef | 0.5 | 0.4 | 0.4 |
| num_envs | 2 | 4 | 8 |
| anneal_lr | True | True | True |
| clip_range_vf | 0.4 | 0.3 | 0.3 |
| update_epochs | 2 | 16 | 16 |
| num_minibatches | 8 | 4 | 4 |

