# OpenReview forum: "Divide and Conquer: Provably Unveiling the Pareto Front with Multi-Objective Reinforcement Learning"
_ICLR.cc/2024/Conference — Submitted to ICLR 2024_

### Official Review · Reviewer_uiiy · 2023-10-18

**Soundness:** 2 fair
**Presentation:** 2 fair
**Contribution:** 2 fair
**Rating:** 3
**Confidence:** 3

**Summary:**

In this paper, the authors propose a new algorithm for multi-objective reinforcement learning. The main idea is to construct a Pareto oracle which can be queried iteratively to shrink the search space of Pareto optimal points. The authors provide theoretical analyses of the convergence properties of the proposed algorithm and experiment with three standard multi-objective optimization environments.

**Strengths:**

1. Multi-objective reinforcement learning is an important problem to study. This paper adds a new method to the collection of MORL algorithms. The proposed Iterated pareto Referent Optimization (IPRO) algorithm is interesting in its three phases and is relevant to the community.

2. I appreciate the authors providing theoretical analyses of IPRO, though I have some questions about their presentations. Please see my questions in the Questions section.

**Weaknesses:**

1. There are some important baselines that need to be included, for example [1] and [2]. Furthermore, comparisons with GPI-LS did not demonstrate any improvements. I think that the authors should provide more discussion on this. It is hard to see why one would want to use IPRO over an existing method.

[1]: Yang, Runzhe, Xingyuan Sun, and Karthik Narasimhan. "A generalized algorithm for multi-objective reinforcement learning and policy adaptation." Advances in neural information processing systems 32 (2019).

[2]: Abels, Axel, et al. "Dynamic weights in multi-objective deep reinforcement learning." International conference on machine learning. PMLR, 2019.

2. The presentation of the paper could use a major improvement. There are several places where definitions are either missing or unclear, making the paper hard to understand. The IPRO algorithm is described in words only in the main paper. I think Algorithms 1 and 2 in the Appendix should be included in the main paper to improve the clarity. Please see my comments in the next sections about other places for clarification.

3. This may be a point I did not understand entirely due to the presentation issue. In Section 3.2, in describing the design of a Pareto oracle, the authors mentioned using GGF-DQN to optimize the objective function in Equation (1). My question is: what is the benefit of utilizing another multi-objective algorithm as a subroutine in IPRO compared to directly using it to optimize the multi-objective function?

**Questions:**

1. In Theorem 4.1, does one need to place some assumptions on the Pareto oracle to claim this result?

2. IPRO needs to find the ideal and nadir vectors for initialization. In practice, even the reduced single-objective RL problem might not be solved to optimality. How does this fact impact the performance of IPRO?

3. In the definition of Achievement scalarizing functions (Section 2), please explain how $s$ becomes $s_r$ depending on $r$.

4. Please define what a target region is. It is first mentioned in the paragraph about Achievement scalarizing functions in Section 2.

5. In the Problem setup (Section 2), please define the expected return value $v^{\pi}$ rigorously.

---

> ### Author Response · Authors · 2023-11-14
> **Response to reviewer 4**
>
> We appreciate the feedback from the reviewer and have addressed the raised questions and suggestions.
>
> **Baselines and presentation**
>
> With respect to the baselines, as mentioned in our general comment, GPI-LS is currently the state-of-the-art method for learning a convex hull. As the Minecart and MO-Reacher have (mostly) convex Pareto fronts, GPI-LS (which explicitly relies on this knowledge) presents a suitable “upper bound” on IPRO which does not make the simplifying assumption that the scalarisation function is linear.
>
> To better convey why one would prefer IPRO over related work, we have introduced Table 1 in the introduction which summarises our contributions. In a nutshell, IPRO addresses a more general setting, discarding the assumption of linearity of the scalarisation function. This allows IPRO to also discover a richer front of Pareto optimal policies as well as tackle different policy classes. We have further attempted to improve the presentation of the paper by removing section 3.2.1 and adding the pseudocode for IPRO to the main text.
>
> **Responses to Raised Questions:**
> 1. Utilising Another Multi-Objective Algorithm as a Subroutine: IPRO serves as a problem-agnostic algorithm, identifying target regions for undiscovered Pareto optimal points and delegating the task to a problem-specific subroutine. This approach allows for flexibility in handling various optimization tasks, extending beyond multi-objective reinforcement learning. The extension of the GGF-DQN algorithm allows for IPRO to be applied to the MORL setting, which we are particularly interested in, but it could indeed be applied more broadly to different domains.
> 2. Theorem 4.1 and Assumptions on Pareto Oracle: While additional assumptions are not explicitly placed on the Pareto oracle, we define the concept of a Pareto oracle ourselves, thus embedding certain desiderata into its definition. One example is that, we ensure that a Pareto oracle always returns an optimal point in its target region, but future work may be able to relax this definition.
> 3. Impact of Single-Objective RL Solution Accuracy on IPRO: While this may indeed happen with certain single-objective methods, this need not be a problem for IPRO. For example, if it is known that the single-objective RL solver has some bounded error, we may simply adjust the nadir and ideal to account for this. On the other hand, we could also “assume” that the method used to generate the nadir and ideal always does so optimally. Then, IPRO will still obtain all points in the resulting bounding box but may fail to retrieve the points outside the bounding box.
> 4. The definition of Achievement scalarising functions: We have updated the definition of $s_\mathbf{r}$ to better explain this. Intuitively, the ASF is instantiated with a referent $\\mathbf{r}$. We have also added a definition of the target region. Formally, the target region given a reference point $\\mathbf{r}$ contains all points $\\mathbf{v}$ such that $\\mathbf{v} \\succeq \\mathbf{r}$. We have added this to the preliminaries.
> 5. Definition $\\mathbf{v}^\\pi$: We have added a rigorous definition of the expected return value $\\mathbf{v}^\\pi$ in the Problem Setup (Section 2) for clarity.
>
> Thank you again for the constructive feedback. Please let us know if you have any additional questions or comments.

---

> > ### Comment · Reviewer_uiiy · 2023-11-22
> > **Thanks for the response**
> >
> > I want to thank the authors for the detailed response. As other reviewers also pointed out, the paper still lacked comparisons with several closely related MORL works. I will maintain my score.

---

### Official Review · Reviewer_3Frz · 2023-11-01

**Soundness:** 3 good
**Presentation:** 2 fair
**Contribution:** 3 good
**Rating:** 6
**Confidence:** 2

**Summary:**

This paper provides a framework for learning the Pareto front in multi-objective MDPs. The framework decomposes the learning problem into a series of single-objective problems, where each problem is solved by a Pareto oracle. Specifically, the IPRO algorithm iteratively proposes reference points to a Pareto oracle and gets new Pareto optimal points which trim sections from the search space. The algorithm is shown to converge to the Pareto front asymptotically. The paper also contains experiments that validate that the algorithm leads to a close approximation to the true Pareto front.

**Strengths:**

- The problem of learning the Pareto front in MOMDPs is general and is important in practice.

- The proposed algorithm is natural and novel.

- The theoretical guarantee looks sound.

**Weaknesses:**

- The presentation is not good enough. The major contribution of this paper is the algorithmic framework IPRO and the guarantee Theorem 4.2. However, the paper introduces IRPO on page 6 while stating Theorem 4.2 on page 8. For example, Section 3.2.1 - 3.2.2 is irrelevant to the main result and should be presented later. Moreover, the absence of the algorithm box of IPRO hinders my understanding.

- See the question part.

**Questions:**

1. How do you select the reference point $l$ from the set of lower points $L$ if there are many candidates?

2. Since the convergence is only asymptotic, how do you show the advantage of your method theoretically? For example, one can optimize a weighted sum utility each time, and the weights are sampled uniformly. This method can also converge to the Pareto front asymptotically.

3. Following 2, is it possible to characterize the convergence rate?

4. What is the big difference between a weak Pareto oracle and an approximate Pareto oracle in practice? Is it only for theoretical rigorousness?

---

> ### Author Response · Authors · 2023-11-14
> **Response to reviewer 3**
>
> We appreciate the insightful questions raised by the reviewer and would like to provide clarifications on the points raised.
>
> **Reference point selection**
>
> The question about selecting the reference point from the set of lower points is addressed in the appendix due to space constraints. In practice, we use a heuristic that chooses the point with the most "open space" (i.e. space not allocated to an excluded subset) dominating it. The intuition is that a reference point with more open space above it is more likely to lead to the discovery of a Pareto optimal point. It is essential to note that IPRO is not reliant on any specific selection method, and this heuristic is employed for practical efficiency.
>
> **Convergence and Utility Function**
>
> We note that while convergence to the true Pareto front is asymptotic, we can guarantee convergence in a finite number of iterations to an arbitrary $\\tau$-Pareto front with $\\tau > 0$. The main benefit of IPRO is that we do not assume that utility is a weighted sum of the objectives, as opposed to Envelope, PGMORL, GPI-LS, and the vast majority of work in MORL. Rather, we allow for any nonlinear utility function. It has been proven that in the case of deterministic policies, the resulting Pareto front may be non-convex and hence methods which rely on linear scalarisation are inherently incapable of retrieving the full front. IPRO in contrast can find all points. We highlight this now in the introduction by adding a table which summarises the contributions of our work compared to relevant related work.
>
> **Convergence rate**
>
> While this is an interesting question, it is not straightforward as the convergence of IPRO is also dependent on the distribution of points on the Pareto front. It may be possible to characterise the behaviour of IPRO in the worst-case distribution of points along a Pareto front. We do not have any concrete results along this line but consider it an interesting topic for future work.
>
> **Pareto oracle definitions**
>
> The distinction between a weak Pareto oracle and an approximate Pareto oracle is primarily for theoretical rigour. However, both types of Pareto oracles may lead to different practical implementations. In the appendix, we describe the implementation of an approximate Pareto oracle using constrained MDPs, which would not be feasible with weak Pareto oracles.
>
> Thank you again for the detailed feedback. If you have any further questions, we would be happy to answer them.

---

### Official Review · Reviewer_iyma · 2023-11-05

**Soundness:** 3 good
**Presentation:** 2 fair
**Contribution:** 2 fair
**Rating:** 3
**Confidence:** 3

**Summary:**

This paper proposes a way to discover the Pareto front of Multi-objective RL (MORL) problems. The authors use the augmented Chebyshev scalarisation function as the achievement scalarising function (ASF), converting a multi-objective problem into a single-objective one. They introduced the Pareto oracles (the weak and the approximate ones) into MORL. With the Pareto oracles in hand, they propose Iterated Pareto Referent Optimization (IPRO), which learns the Pareto front in 3 phases: (i) The main loop consists of iteratively updating the Pareto front by interacting with the Pareto oracle. (ii) The return value from the Pareto oracle can be used to update the Pareto front while maintaining the lower set L and upper set U. Finally, they theoretically give an upper bound on the approximation error and a guarantee to converge to a \tau-Pareto front. The proposed algorithm is then evaluated on multiple benchmark MORL environments, including DST, Minecraft and MO-Reacher.

**Strengths:**

- This paper introduces a new concept called Pareto oracle into MORL, which can be implemented with modifications of single-objective RL algorithms, and thereby proposes IPRO, a new MORL framework that iteratively finds the Pareto optimal policies with the help of Pareto oracles.

- This paper then provides upper bounds on the true approximation error at each time step and gives a convergence guarantee of their proposed algorithm IPRO.

**Weaknesses:**

- My main concern is that the proposed IPRO framework completely ignores one core problem of RL (and hence MORL) -- sample efficiency. Specifically, IPRO completely abstracts away the "learning from online interactions" aspect of RL through the use of Pareto oracles, which are assumed to be capable of directly returning a Pareto optimal policy given any reference point. However, this is actually the fundamental challenge of RL (and surely MORL as well). If the complexity of "learning from online interactions" is completely encapsulated into an oracle, then the problem would simply degenerate to a typical Multi-Objective Optimization (MOO) problem and then there is no need for reinforcement learning (That also manifests why the experimental results are all shown in terms of iterations, not in number of samples). With that said, the IPRO appears more like just an MOO algorithm rather than an MORL method (as it is agnostic to the specific problem structure of RL).

- Several algorithmic components are stated without much explanation and hence rather confusing. For example:
    - Regarding the practical implementation of a Pareto oracle, a Pareto oracle could be achieved by taking the augmented Chebyshev scalarisation function as the objective under any off-the-shelf RL algorithm (e.g., DQN, A2C, etc). Notably, the augmented Chebyshev function depends not only on the reference point but also  $\lambda$ and $\rho$, which are the weight vector and the augmentation coefficient that determine the required improvement of each dimension. The choices of $\lambda$ and $\rho$ are critical in determining which Pareto optimal solution we are looking for (similar to the preference vector in linear scalarization). However, in IPRO, it is not explained (either in the main text of the pseudo code in Appendix) how these parameters are determined in each iteration.
    - IPRO, how shall we construct the sets $L$ and $U$?
    - What has been done in the Pruning phase? And how does Pruning uncover additional Pareto optimal solutions?
    - How is the memory-based policy implemented? Is there any special neural network architecture involved?

- The experimental results do not appear strong enough to demonstrate the performance of IPRO. Several MORL benchmark methods are missing in the experiments. Just to name a few:
    - PGMORL (Xu et al., ICML 2020) uses an evolutionary approach to search for the Pareto front and shall be a good baseline for IPRO.
    - Envelope Q-learning (Yang et al., NeurIPS 2019) and (Abels et al., ICML 2019) provide single-network solutions to approach the convex coverage set.
    - More recently, (Basaklar et al., ICLR 2023) and (Hung et al., ICLR 2023) also provide single-network solutions to MORL.

Moreover, the experimental results are all reported in terms of iteration, which does not reflect the actual sample efficiency of the algorithms.

- The presentation could be improved in several places. For example, in Section 2 and Section 3, while the preliminaries and definitions are mostly fairly standard definitions and concepts in MORL, I do find the description to be somewhat lengthy and hence a bit hard to read.



[References]
1. Jie Xu, Yunsheng Tian, Pingchuan Ma, Daniela Rus, Shinjiro Sueda, and Wojciech Matusik, “Prediction-guided multi-objective reinforcement learning for continuous robot control,” ICML 2020.
2. Runzhe Yang, Xingyuan Sun, and Karthik Narasimhan, “A generalized algorithm for multi-objective reinforcement learning and policy adaptation,” NeurIPS 2019.
3. Axel Abels, Diederik Roijers, Tom Lenaerts, Ann Now´e, and Denis Steckelmacher, “Dynamic weights in multi-objective deep reinforcement learning,” ICML 2019
4. Toygun Basaklar, Suat Gumussoy, Umit Ogras, “PD-MORL: Preference-Driven Multi-Objective Reinforcement Learning Algorithm,” ICLR 2023.
5. Wei Hung, Bo Kai Huang, Ping-Chun Hsieh, Xi Liu, “Q-Pensieve: Boosting Sample Efficiency of Multi-Objective RL Through Memory Sharing of Q-Snapshots,” ICLR 2023.

**Questions:**

Some additional detailed questions:

- What are the definitions of “nadir” $v^n$ and “ideal” vector $v^i$ in Figures 1 and 2? (They seem to be some uniform lower bound and upper bound of all possible return vectors?)

- What is the reason behind the design of a* in eq (3)? Is this design theoretically grounded?

- I notice that in Fig 3(e), the coverage of DQN as Pareto oracle is generally higher than the other two while the hypervolume of DQN as Pareto oracle is the lowest in Fig 3(b). Why is that the case?

- What is the main usage of Section 3.2.1? Is the stochastic stationary policy used in the implementation?

---

> ### Author Response · Authors · 2023-11-14
> **Response to reviewer 2**
>
> We appreciate the detailed feedback from the reviewer and would like to address the raised points.
>
> **Sample Complexity and Learning From Interactions**
>
> We believe the reviewer strikes an important point. Indeed, given our current training setup, we train a new policy in each oracle call. However, it is likely that a single-network approach is also possible with IPRO. As our primary focus was on novel theoretical results in this challenging setting, we did not yet opt for this approach but consider it a promising direction for future work.
>
> As suggested, IPRO is indeed problem-agnostic and can thus be applied for general multi-objective problems. We are primarily interested in learning a Pareto front in MOMDPs where no explicit model is given, hence necessitating the use of RL as a subroutine. We plan to evaluate IPRO in related domains, such as optimisation and pathfinding, in the future.
>
> **$\lambda$ and $\rho$ parameters**
>
> We note that $\lambda$ is a weight vector that is supplied by IPRO and is computed as $(\mathbf{v}^\text{i} - \mathbf{r})^{-1}$ where $\mathbf{v}^\text{i}$ is the ideal and $\mathbf{r}$ the referent. This normalises the improvement of a vector $\mathbf{v}$ relative to the referent $\mathbf{r}$ by the theoretical maximum improvement at the ideal $\mathbf{v}^\text{i}$. The purpose of this normalisation is to maintain a balanced scale across all objectives, preventing the dominance of one objective over another. $\rho$, on the other hand, is a hyperparameter which we tune.
>
> **$L$ and $U$ sets**
>
> The construction of the sets $L$ and $U$ is now detailed more explicitly in Appendix B1. Intuitively, this is a sequential process that starts from the initial $L$ and $U$ sets. Then, every time a new point is added to the Pareto front, we can compute the added “corners” of the boundaries of the dominated and dominating sets. These new corners are then added to $L$ and $U$. For a visualisation, we refer to Figure 2. Pseudocode is also presented in Algorithm 2.
>
> **Brief answers to the remaining questions:**
> 1. The use of pruning: Pruning does not uncover additional points, but rather rejects points which are not Pareto optimal but were added during IPRO’s execution. This step may be safely omitted without changing anything except possibly returning an unnecessarily large set.
> 2. Memory-based policy implementation: We concatenate the memory (which is the accrued reward until the given timestep) to the observation and feed this into the neural network. This induces what is known in the MORL literature as an “augmented state space” and is a common approach.
> 3. Definitions of “nadir” and “ideal” vector: The nadir $\mathbf{v}^\text{n}$ is defined as the “greatest lower bound” while the ideal $\mathbf{v}^\text{i}$ is defined as the “least upper bound”. To give an example, let $PF = \\{(1, 2), (2, 0)\\}$ then the nadir is $(1, 0)$ while the ideal is $(2, 2)$. These quantities are defined in the preliminaries.
> 4. The design of $a^\ast$ in eq (3): Intuitively, this definition “mimics” the optimisation objective of the agent where it aims to optimise its scalarisation function applied to the expected returns of the policy. While this update does not exactly represent this objective, it is common in the MORL literature and works well in practice.
> 5. The discrepancy between coverage and hypervolume: The DQN agent fails to find a Pareto optimal point in some target region and then incorrectly excludes the search space in the region from consideration. As such, it has a higher coverage while still resulting in a lower hypervolume. For future work, we plan to extend IPRO to incorporate uncertainty over which sections of the search space may still contain optimal points rather than the current approach of including/excluding search space.
> 6. Usage of Section 3.2.1: We intended to inform the reader that the concept of a Pareto oracle is feasible to implement and comes with theoretical grounding for specific policy classes. We did not use it in the remainder of the text. However, most reviewers agreed that this section was unnecessary and complicated the understanding of our paper. We have therefore moved this section to the appendix and added the pseudocode of IPRO to the main text instead.
>
> Thank you again for the detailed feedback. If you have any further questions, we would be happy to answer them.

---

> ### Comment · Reviewer_iyma · 2023-11-22
>
> Thank the authors for the detailed response. The algorithmic details have been made more transparent in several places.
> That said, regarding the sample efficiency and the comparison with the MORL benchmark methods, my concerns still nevertheless remain, as also pointed out by other reviewers. Therefore, I tend to maintain my original rating.

---

### Official Review · Reviewer_WEZT · 2023-11-10

**Soundness:** 2 fair
**Presentation:** 2 fair
**Contribution:** 2 fair
**Rating:** 3
**Confidence:** 4

**Summary:**

This paper proposes IRPO, which transforms multi-objective optimization problems into a series of single-objective optimization problems. It iteratively selects reference points and simplifies the search space based on the solutions optimized by RL. The effectiveness of the method is demonstrated both experimentally and theoretically in this paper.

**Strengths:**

1, The idea presented in this paper, as far as I know, is innovative. The approach of decomposing the problem from the perspective of hypervolume and solving it gradually is inspiring to me. I believe this paper has the potential to become a work with long-term impact.

**Weaknesses:**

1. Despite the novelty of the idea, the major concern is that the experimental evidence in the paper is not sufficient to demonstrate the advantages of the method compared to other MORL algorithms. First, the benchmarks used are relatively simple, and there are no experiments on complex multi-objective tasks, such as the classic MO-MuJoCo tasks. Additionally, there is a lack of comparisons with relevant MORL baselines, such as Envelope [1], PGMORL [2], and Q-Pensieve [3].
2. Further improvements are needed in the presentation of the paper. For instance,  the definitions in Chapter 4 could be introduced in Chapters 2 and 3, rather than gradually unfolding them within later sections.
3. I strongly recommend the authors to include pseudocode in the main text.


---
[1]: A Generalized Algorithm for Multi-Objective Reinforcement Learning and Policy Adaptation
[2]: Prediction-Guided Multi-Objective Reinforcement Learning for Continuous Robot Control
[3]: Q-Pensieve: Boosting Sample Efficiency of Multi-Objective RL Through Memory Sharing of Q-Snapshots

**Questions:**

1. What are the advantages of IRPO compared to directly using RL for multi-objective optimization? For example, I could scalarize the multi-objective rewards directly and optimize them. I could use Envelope [1] to train one policy to solve all tasks, or get a set of policies through PGMORL [2].
2. Could the authors provide a performance comparison between IRPO and other MORL algorithms on MO-MuJoCo?
3. What do "occupancy measure" and "occupancies" mean in Section 3.2.1? An introduction is lacking.
4. How long does the RL optimization process before pruning take? More detailed training specifics are needed.

If the authors can address my concerns, I am willing to increase my score.

---

> ### Author Response · Authors · 2023-11-14
> **Response to reviewer 1**
>
> We appreciate the positive feedback from the reviewer regarding the contributions of our paper. Below, we address the main remarks and also go over the detailed questions.
>
> **Advantages of IPRO**
>
> In the vast majority of MORL, exemplified by references [1] (envelope) and [2] (PG-MORL), a simplifying assumption is made that decision-makers have linear preferences, resulting in a convex Pareto front and the sufficiency of deterministic policies. By forgoing such assumptions, IPRO gains a unique advantage in addressing more general settings. To support this claim, we focus on deterministic memory-based policies, a class where the Pareto front may exhibit non-convexity. Indeed, the assumption of linear preferences in the Deep Sea Treasure environment leads to only two of the ten Pareto optimal policies being retrievable. Critically, IPRO does not assume linear preferences and is thus able to find all Pareto optimal solutions. We present a summarising table in the introduction, highlighting IPRO's advantages compared to related work.
>
> **Experimental Results**
>
> Addressing the concern about experimental results, we conducted an extended hyperparameter search, resulting in significant improvements across all experiments. In addition, it is worth noting that the Reacher benchmark is a MuJoCo environment and that the remaining multi-objective MuJoCo environments in MO-Gymnasium have continuous actions. While IPRO is agnostic to the underlying problem and thus could handle continuous actions, this would require significant additional contributions as there has been limited work tackling such complex continuous control settings with nonlinear scalarisations. Finally, we did not compare against Envelope, PG-MORL and Q-Pensieve as GPI-LS has been reported to reach state-of-the-art results on the chosen benchmarks.
>
> **Presentation**
>
> As stated in our general comment, we have moved Section 3.2.1 to the appendix and added pseudocode for IPRO to the main text. We believe that this improves the presentation.
>
> **Occupancy measure in section 3.2.1**
>
> We moved this section to the appendix to avoid confusion. To briefly clarify, an occupancy measure intuitively denotes how frequently a specific state is visited when employing a particular policy. This metric is commonly used in concave utility RL and for deriving theoretical results in classical RL settings.
>
> **Training details and pruning**
>
> Each iteration of the Pareto oracle runs for a fixed number of steps, which is reported in the appendix. After the entire execution of IPRO has concluded, we perform one pruning step to remove weakly Pareto dominant solutions. While not strictly necessary, this step avoids presenting unnecessary solutions to decision-makers. To enhance clarity, we have removed the mention of pruning from the main text, addressing this aspect instead in the appendix. Additionally, we've expanded the appendix to provide additional details regarding the algorithm (Appendix B1) and training procedure (Appendix D).
>
> Thank you again for the constructive feedback. Please let us know if you have any additional questions or comments.

---

### Author Response · Authors · 2023-11-14
**General response to comments**

We appreciate the constructive feedback provided by the reviewers and are pleased to observe their recognition of the novelty and theoretical underpinnings of our algorithm. Taking into account the collective suggestions, we have made substantial revisions to enhance the paper.


**Experimental results**

Responding to the shared observation across all reviewers regarding the limited experimental evidence, we conducted an extended hyperparameter search across a broader parameter range. Consequently, we have significantly improved the reported results. The updated paper now reflects these enhancements. Notably, in the Deep Sea Treasure scenario, our algorithm approaches the theoretical optimum. This is important as MORL algorithms which assume linear preferences cannot solve this environment due to its non-convex Pareto front. Moreover, in the Minecart environment, we are now comparable to GPI-LS. This is particularly promising as GPI-LS relies on the knowledge of the convex nature of the Pareto front in Minecart, a prerequisite not assumed by IPRO. Therefore, IPRO achieves comparable results with relaxed domain knowledge. Finally, we have refined our results in the Reacher scenario, with policy gradient oracles now demonstrating performance levels between PCN and GPI-LS. Here too, the Pareto front is mostly convex, which makes GPI-LS a suitable upper bound.

**Clarity of Presentation**

Recognising the reviewers' concerns about the clarity of our presentation, we have taken several measures to address this issue. In the introduction, we have incorporated a table summarising our contributions compared to related work, providing a better contextualisation of our algorithm within the literature. Furthermore, we have eliminated Section 3.2.1, which originally aimed to convey the feasibility of implementing a theoretically sound Pareto oracle. As this result is not further used in the paper, we have relocated it to the appendix. Finally, to enhance the clarity of our algorithm's presentation, we have included pseudocode of IPRO in the main text.

---

### Meta-Review · Area_Chair_gE7c · 2023-12-12

**Metareview:**

This paper proposes a novel method IPRO for learning the Pareto front in multi-objective MDPs and studies its theoretical convergence guarantee.  Though the theories and numerical results seem sound, most reviewers agree that this paper lacks a thorough comparison with other baselines and the study of the sample efficiency of IPRO. Compared with a single baseline cannot convince the reader of the effectiveness of IPRO. As pointed out by Reviewer iyma, the question of sample efficiency may hurt this paper's contribution severely and more experimental studies are required.
In the current form, I tend to recommend rejection.

**Justification For Why Not Higher Score:**

It lacks comparisons with some important baselines (see Reviewer uiiy) and the experiments on sample efficiency.

**Justification For Why Not Lower Score:**

N/A

---

### Decision · Program_Chairs · 2024-01-16

Reject